# Integrating genotypes and phenotypes improves long-term forecasts of seasonal influenza A/H3N2 evolution

John Huddleston[1,2]\*, John R Barnes[3], Thomas Rowe[3], Xiyan Xu[3], Rebecca Kondor[3], David E Wentworth[3], Lynne Whittaker[4], Burcu Ermetal[4], Rodney Stuart Daniels[4], John W McCauley[4], Seiichiro Fujisaki[5], Kazuya Nakamura[5], Noriko Kishida[5], Shinji Watanabe[5], Hideki Hasegawa[5], Ian Barr[6], Kanta Subbarao[6], Pierre Barrat-Charlaix[7,8], Richard A Neher[7,8], Trevor Bedford[1]\*

[1]Vaccine and Infectious Disease Division, Fred Hutchinson Cancer Research Center, Seattle, United States; [2]Molecular and Cell Biology Program, University of Washington, Seattle, United States; [3]Virology Surveillance and Diagnosis Branch, Influenza Division, National Center for Immunization and Respiratory Diseases (NCIRD), Centers for Disease Control and Prevention (CDC), Atlanta, United States; [4]WHO Collaborating Centre for Reference and Research on Influenza, Crick Worldwide Influenza Centre, The Francis Crick Institute, London, United Kingdom; [5]Influenza Virus Research Center, National Institute of Infectious Diseases, Tokyo, Japan; [6]The WHO Collaborating Centre for Reference and Research on Influenza, The Peter Doherty Institute for Infection and Immunity, Department of Microbiology and Immunology, The University of Melbourne, The Peter Doherty Institute for Infection and Immunity, Melbourne, Australia; [7]Biozentrum, University of Basel, Basel, Switzerland; [8]Swiss Institute of Bioinformatics, Basel, Switzerland

**\*For correspondence:**
jlhudd@uw.edu (JH);
tbedford@fhcrc.org (TB)

**Competing interests:** The authors declare that no competing interests exist.

**Abstract** Seasonal influenza virus A/H3N2 is a major cause of death globally. Vaccination remains the most effective preventative. Rapid mutation of hemagglutinin allows viruses to escape adaptive immunity. This antigenic drift necessitates regular vaccine updates. Effective vaccine strains need to represent H3N2 populations circulating one year after strain selection. Experts select strains based on experimental measurements of antigenic drift and predictions made by models from hemagglutinin sequences. We developed a novel influenza forecasting framework that integrates phenotypic measures of antigenic drift and functional constraint with previously published sequence-only fitness estimates. Forecasts informed by phenotypic measures of antigenic drift consistently outperformed previous sequence-only estimates, while sequence-only estimates of functional constraint surpassed more comprehensive experimentally-informed estimates. Importantly, the best models integrated estimates of both functional constraint and either antigenic drift phenotypes or recent population growth.

## Introduction

Seasonal influenza virus infects 5–15% of the global population every year causing an estimated 250,000 to 500,000 deaths annually with the majority of infections caused by influenza A/H3N2 (*World Health Organization, 2014*). Vaccination remains the most effective public health response available. However, frequent viral mutation results in viruses that escape previously acquired human immunity. The World Health Organization (WHO) Global Influenza Surveillance and Response System

**eLife digest** Vaccination is the best protection against seasonal flu. It teaches the immune system what the flu virus looks like, preparing it to fight off an infection. But the flu virus changes its molecular appearance every year, escaping the immune defences learnt the year before. So, every year, the vaccine needs updating. Since it takes almost a year to design and make a new flu vaccine, researchers need to be able to predict what flu viruses will look like in the future. Currently, this prediction relies on experiments that assess the molecular appearance of flu viruses, a complex and slow approach.

One alternative is to examine the virus's genetic code. Mathematical models try to predict which genetic changes might alter the appearance of a flu virus, saving the cost of performing specialised experiments. Recent research has shown that these models can make good predictions, but including experimental measures of the virus' appearance could improve them even further. This could help the model to work out which genetic changes are likely to be beneficial to the virus, and which are not.

To find out whether experimental data improves model predictions, Huddleston et al. designed a new forecasting tool which used 25 years of historical data from past flu seasons. Each forecast predicted what the virus population might look like the next year using the previous year's genetic code, experimental data, or both. Huddleston et al. then compared the predictions with the historical data to find the most useful data types. This showed that the best predictions combined changes from the virus's genetic code with experimental measures of its appearance.

This new forecasting tool is open source, allowing teams across the world to start using it to improve their predictions straight away. Seasonal flu infects between 5 and 15% of the world's population every year, causing between quarter of a million and half a million deaths. Better predictions could lead to better flu vaccines and fewer illnesses and deaths.

(GISRS) monitors influenza evolution by sampling currently circulating viruses, or strains, and analyzing these strains with genome sequencing and serological assays. The WHO GISRS uses these data to select vaccine viruses that should best represent circulating viruses in the next influenza season. However, because the process of vaccine development and distribution requires several months to complete, optimal vaccine design requires an accurate prediction of which viruses will predominate approximately one year after vaccine viruses are selected.

Historically, the effectiveness of the H3N2 vaccine component has been much lower than the other seasonal influenza subtypes. For example, H3N2's mean vaccine effectiveness from 2004 to 2015 was 33% compared to 61% for H1N1pdm and 54% for influenza B viruses (*Belongia et al., 2016*). Multiple factors can reduce vaccine effectiveness including selection of a vaccine strain that is not antigenically representative of future populations (*Belongia et al., 2016*; *Gouma et al., 2020*) and adaptations of the selected strain to egg-passaging during vaccine production that alter the antigenicity of the resulting vaccine component (*Zost et al., 2017*). Even when vaccine strains are well-matched antigenically, they may fail to induce a strong immune response due to previous infection history of vaccine recipients (*Cobey et al., 2018*). While all of these factors must be addressed to increase vaccine effectiveness, substantial effort has focused on the selection of the most representative strain for the next season's vaccine.

Current vaccine predictions focus on the hemagglutinin (HA) protein, which acts as the primary target of human immunity. Until recently, the hemagglutination inhibition (HI) assay has been the primary experimental measure of antigenic cross-reactivity between pairs of circulating viruses (*Hirst, 1943*). Most modern H3N2 strains carry a glycosylation motif that reduces their binding efficiency in HI assays (*Chambers et al., 2015*; *Zost et al., 2017*), prompting the increased use of virus neutralization assays including the neutralization-based focus reduction assay (FRA) (*Okuno et al., 1990*). Together, these two assays are the gold standard in virus antigenic characterizations for vaccine strain selection, but they are laborious and low-throughput compared to genome sequencing (*Wood et al., 2012*). As a result, researchers have developed computational methods to predict influenza evolution from sequence data alone (*Luksza and Lässig, 2014*; *Steinbrück et al., 2014*; *Neher et al., 2014*).

Despite the promise of these sequence-only models, they explicitly omit experimental measurements of antigenic or functional phenotypes. Recent developments in computational methods and influenza virology have made it feasible to integrate these important metrics of influenza fitness into a single predictive model. For example, phenotypic measurements of antigenic drift are now accessible through phylogenetic models (*Neher et al., 2016*) and functional phenotypes for HA are available from deep mutational scanning (DMS) experiments (*Lee et al., 2018*). We describe an approach to integrate previously disparate sequence-only models of influenza evolution with high-quality experimental measurements of antigenic drift and functional constraint.

The influenza community has long recognized the importance of incorporating HI phenotypes and other experimental measurements of viral phenotypes with existing forecasting methods to inform the vaccine design process (*Gandon et al., 2016*; *Morris et al., 2018*; *Lässig et al., 2017*). Although several distinct efforts have made progress in using HI phenotypes to evaluate the evolution of seasonal influenza (*Steinbrück et al., 2014*; *Neher et al., 2016*), published methods stop short of developing a complete forecasting framework wherein the evolutionary contribution of HI phenotypes can be compared and contrasted with new and existing fitness metrics. However, unpublished work by *Luksza and Lässig, 2014* to the WHO GISRS network incorporates antigenic phenotypes into fitness-based predictions (*Morris et al., 2018*; M Łuksza, personal communication, June 2020). Here, we provide an open source framework for forecasting the genetic composition of future seasonal influenza populations using genotypic and phenotypic fitness estimates. We apply this framework to HA sequence data shared via the GISAID EpiFlu database (*Shu and McCauley, 2017*) and to HI and FRA titer data shared by WHO GISRS Collaborating Centers in London, Melbourne, Atlanta and Tokyo. We systematically compare potential predictors and show that HI phenotypes enable more accurate long-term forecasts of H3N2 populations compared to previous metrics based on epitope mutations alone. We also find that composite models based on phenotypic measures of antigenic drift and genotypic measures of functional constraint consistently outperform any fitness models based on individual genotypic or phenotypic metrics.

## Results

### A distance-based model of seasonal influenza evolution

We developed a framework to forecast seasonal influenza evolution inspired by the Malthusian growth fitness model of *Luksza and Lässig, 2014*. As with this original model, we forecasted the frequencies of viral populations one year in advance by applying to each virus strain an exponential growth factor scaled by an estimate of the strain's fitness (*Figure 1* and *Equation 1*). *Luksza and Lässig, 2014* measured model performance by identifying clades – groups of strains that all share a recent common ancestor – and comparing observed and estimated future clade frequencies. However, as clade definitions are inherently unstable between seasons, we evaluated our models by comparing the genetic composition of observed and estimated future populations with the earth mover's distance metric. The earth mover's distance calculates the minimum distance between two populations, given the frequency of each individual within a population and a pairwise 'ground distance' between individuals (*Rubner et al., 1998*). We defined distinct amino acid haplotypes as individuals in our observed and estimated future populations. For frequencies of individuals, we used the observed frequencies of haplotypes in the future and our model's estimated frequencies. We calculated the ground distance between individuals as the Hamming distance between haplotypes. With this implementation, more accurate projections of the future population's composition produce smaller earth mover's distances between the observed and estimated future (*Figure 1*).

We estimated viral fitness with biologically-informed metrics including those originally defined by *Luksza and Lässig, 2014* of epitope antigenic novelty and mutational load (non-epitope mutations) as well as four more recent metrics including hemagglutination inhibition (HI) antigenic novelty (*Neher et al., 2016*), deep mutational scanning (DMS) mutational effects (*Lee et al., 2018*), local branching index (LBI) (*Neher et al., 2014*), and change in clade frequency over time (delta frequency) (*Table 1*). All of these metrics except for HI antigenic novelty and DMS mutational effects rely only on HA sequences. The antigenic novelty metrics estimate how antigenically distinct each strain at time $t$ is from previously circulating strains based on either genetic distance at epitope sites or $\log_2$ titer distance from HI measurements. Increased antigenic drift relative to previously

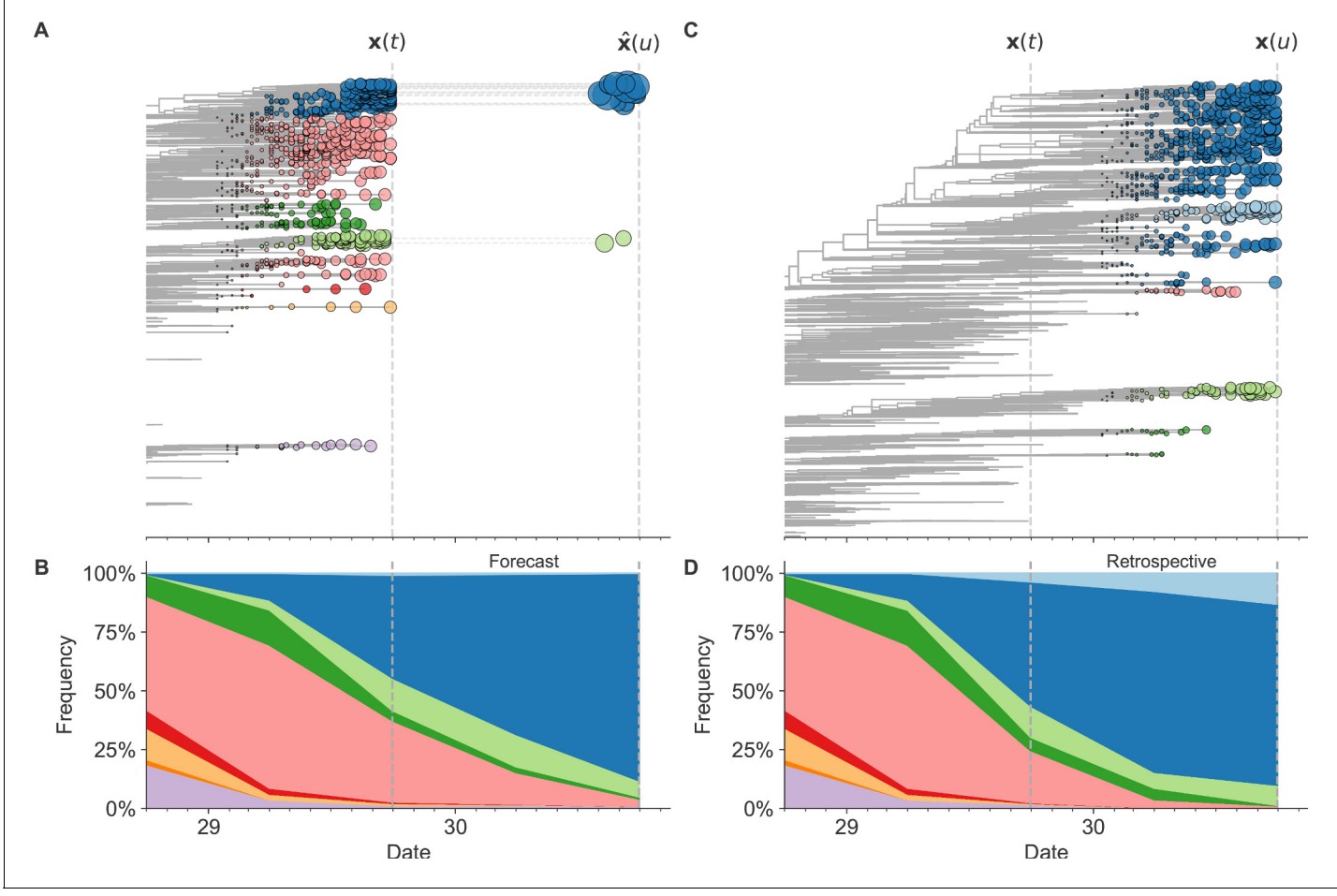

**Figure 1.** Schematic representation of the fitness model for simulated H3N2-like populations wherein the fitness of strains at timepoint $t$ determines the estimated frequency of strains with similar sequences one year in the future at timepoint $u$. Strains are colored by their amino acid sequence composition such that genetically similar strains have similar colors (Materials and methods). (A) Strains at timepoint $t$, $\mathbf{x}(t)$, are shown in their phylogenetic context and sized by their frequency at that timepoint. The estimated future population at timepoint $u$, $\hat{\mathbf{x}}(u)$, is projected to the right with strains scaled in size by their projected frequency based on the known fitness of each simulated strain. (B) The frequency trajectories of strains at timepoint $t$ to $u$ represent the predicted the growth of the dark blue strains to the detriment of the pink strains. (C) Strains at timepoint $u$, $\mathbf{x}(u)$, are shown in the corresponding phylogeny for that timepoint and scaled by their frequency at that time. (D) The observed frequency trajectories of strains at timepoint $u$ broadly recapitulate the model's forecasts while also revealing increased diversity of sequences at the future timepoint that the model could not anticipate, e.g. the emergence of the light blue cluster from within the successful dark blue cluster. Model coefficients minimize the earth mover's distance between amino acid sequences in the observed, $\mathbf{x}(u)$, and estimated, $\hat{\mathbf{x}}(u)$, future populations across all training windows.

circulating strains is expected to correspond to increased viral fitness. Mutational load estimates functional constraint by measuring the number of putatively deleterious mutations that have accumulated in each strain since their ancestor in the previous season. DMS mutational effects provide a more comprehensive biophysical model of functional constraint by measuring the beneficial or deleterious effect of each possible single amino acid mutation in HA from the background of a previous vaccine strain, A/Perth/16/2009. The growth metrics estimate how successful populations of strains have been in the last six months based on either rapid branching in the phylogeny (LBI) or the change in clade frequencies over time (delta frequency).

We fit models for individual fitness metrics and combinations of metrics that we anticipated would be mutually beneficial. For each model, we learned coefficient(s) that minimized the earth mover's distance between HA amino acid sequences from the observed population one year in the future and the estimated population produced by the fitness model (*Equation 2*). We evaluated model performance with time-series cross-validation such that better models reduced the earth mover's distance to the future on validation or test data. The earth mover's distance to the future can

**Table 1.** Summary of models used with simulated and natural populations.
Models are labeled by the type of population they were applied to, the type of data they were based on, and the component of influenza fitness they represent.

| Model | Populations | Data type | Fitness category | Originally implemented by |
|---|---|---|---|---|
| true fitness | simulated | simulated populations | positive control | this study |
| naive | simulated, natural | HA sequences | negative control | this study |
| epitope antigenic novelty | simulated, natural | HA sequences | antigenic drift | *Luksza and Lässig, 2014* |
| epitope ancestor | simulated, natural | HA sequences | antigenic drift | *Luksza and Lässig, 2014* |
| HI antigenic novelty | natural | serological assays | antigenic drift | this study |
| mutational load | simulated, natural | HA sequences | functional constraint | *Luksza and Lässig, 2014* |
| deep mutational scanning (DMS) mutational effects | natural | DMS assays | functional constraint | *Lee et al., 2018* |
| local branching index (LBI) | simulated, natural | HA sequences | clade growth | *Neher et al., 2014* |
| delta frequency | simulated, natural | HA sequences | clade growth | this study |

never be zero, because each model makes predictions based on sequences available at the time of prediction and cannot account for new mutations that occur during the prediction interval. We calculated the lower bound for each model's performance as the optimal distance to the future possible given the current sequences at each timepoint. As an additional reference, we evaluated the performance of a 'naive' model that predicted the future population would be identical to the current population. We expected that the best models would consistently outperform the naive model and perform as close as possible to the lower bound.

## Models accurately forecast evolution of simulated H3N2-like viruses

The long-term evolution of influenza H3N2 hemagglutinin has been previously described as a balance between positive selection for substitutions that enable escape from adaptive immunity by modifying existing epitopes and purifying selection on domains that are required to maintain the protein's primary functions of binding and membrane fusion (*Bush et al., 1999*; *Neher, 2013*; *Luksza and Lässig, 2014*; *Koelle and Rasmussen, 2015*). To test the ability of our models to accurately detect these evolutionary patterns under controlled conditions, we simulated the long-term evolution of H3N2-like viruses under positive and purifying selection for 40 years (Materials and methods, *Figure 2*). These selective constraints produced phylogenetic structures and accumulation of epitope and non-epitope mutations that were consistent with phylogenies of natural H3N2 HA (*Figure 3, Tables 2* and *3*). We fit models to these simulated populations using all sequence-only fitness metrics. As a positive control for our model framework, we also fit a model based on the true fitness of each strain as measured by the simulator.

We hypothesized that fitness metrics associated with viral success such as true fitness, epitope antigenic novelty, LBI, and delta frequency would be assigned positive coefficients, while metrics associated with fitness penalties, like mutational load, would receive negative coefficients. We reasoned that both LBI and delta frequency would individually outperform the mechanistic metrics as both of these growth metrics estimate recent clade success regardless of the mechanistic basis for that success. Correspondingly, we expected that a composite model of epitope antigenic novelty and mutational load would perform as well as or better than the growth metrics, as this model would include both primary fitness constraints acting on our simulated populations.

As expected, the true fitness model outperformed all other models, estimating a future population within 6.82 ± 1.52 amino acids (AAs) of the observed future and surpassing the naive model in 32 (97%) of 33 timepoints (*Figure 4*, *Table 4*). Although the true fitness model performed better than the naive model's average distance of 8.97 ± 1.35 AAs, it did not reach the closest possible distance between populations of 4.57 ± 0.61 AAs. With the exception of epitope antigenic novelty, all biologically-informed models consistently outperformed the naive model (*Figure 5*, *Table 4*). LBI was the best of these models, with a distance to the future of 7.57 ± 1.85 AAs. This result is consistent with the fact that the LBI is a correlate of fitness in models of rapidly adapting populations

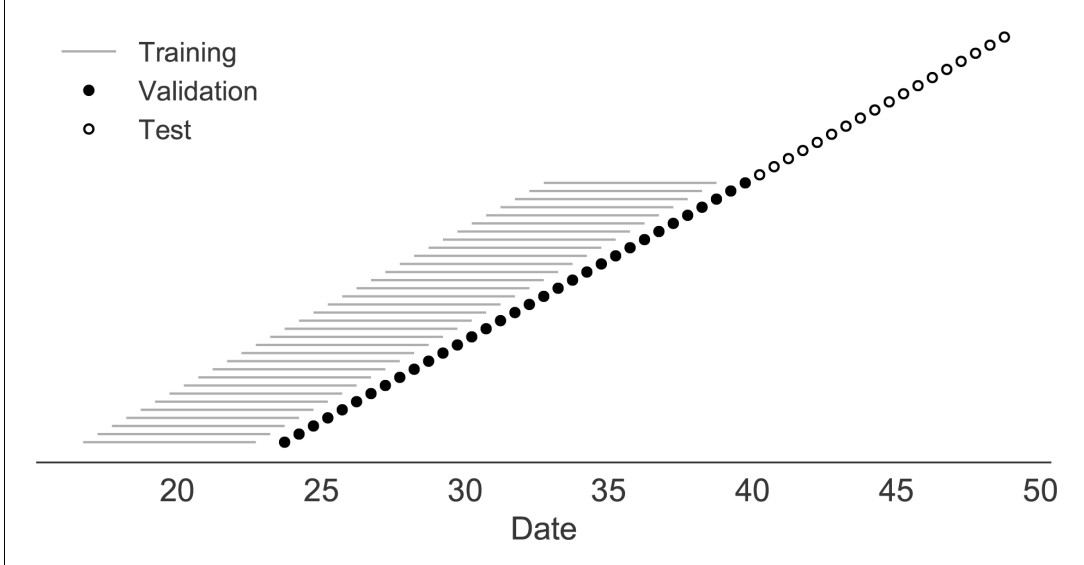

**Figure 2.** Time-series cross-validation scheme for simulated populations. Models were trained in six-year sliding windows (gray lines) and validated on out-of-sample data from validation timepoints (filled circles). Validation results from 30 years of data were used to iteratively tune model hyperparameters. After fixing hyperparameters, model coefficients were fixed at the mean values across all training windows. Fixed coefficients were applied to 9 years of new out-of-sample test data (open circles) to estimate true forecast errors.

(*Neher et al., 2014*). Indeed, both growth-based models received positive coefficients and outperformed the mechanistic models. The mutational load metric received a consistently negative coefficient with an average distance of 8.27 ± 1.35 AAs.

Surprisingly, the composite model of epitope antigenic novelty and mutational load did not perform better than the individual mutational load model (*Figure 5—figure supplement 1*). The antigenic novelty fitness metric assumes that antigenic drift is driven by nonlinear effects of previous host exposure (*Luksza and Lässig, 2014*) that are not explicitly present in our simulations. To understand whether positive selection at epitope sites might be better represented by a linear model, we fit an additional model based on an 'epitope ancestor' metric that counted the number of epitope mutations since each strain's ancestor in the previous season. This linear fitness metric slightly outperformed the antigenic novelty metric (*Table 4*). Importantly, a composite model of the epitope ancestor and mutational load metrics outperformed all other epitope-based models and the individual mutational load model (*Figure 5—figure supplement 1*). From these results, we concluded that our method can accurately estimate the evolution of simulated populations, but that the fitness of simulated strains was dominated by purifying selection and only weakly affected by a linear effect of positive selection at epitope sites.

We hypothesized that a composite model of mutually beneficial metrics could better approximate the true fitness of simulated viruses than models based on individual metrics. To this end, we fit an additional model including the best metrics from the mechanistic and clade growth categories: mutational load and LBI. This composite model outperformed both of its corresponding individual metric models with an average distance to the future of 7.24 ± 1.66 AAs and outperformed the naive model as often as the true fitness metric (*Figure 5*, *Table 4*, *Table 5*). The coefficients for mutational load and LBI remained relatively consistent across all validation timepoints, indicating that these fitness metrics were stable approximations of the simulator's underlying evolutionary processes. This small gain supports our hypothesis that multiple complementary metrics can produce more accurate models.

We validated the best performing model (true fitness) using two metrics that are relevant for practical influenza forecasting and vaccine design efforts. First, we measured the ability of the true fitness model to accurately estimate dynamics of large clades (initial frequency >15%) by comparing observed fold change in clade frequencies, $\log_{10} \frac{x(t+\Delta t)}{x(t)}$ and estimated fold change, $\log_{10} \frac{\hat{x}(t+\Delta t)}{x(t)}$. The model's estimated fold changes correlated well with observed fold changes (Pearson's $R^2 = 0.52$,

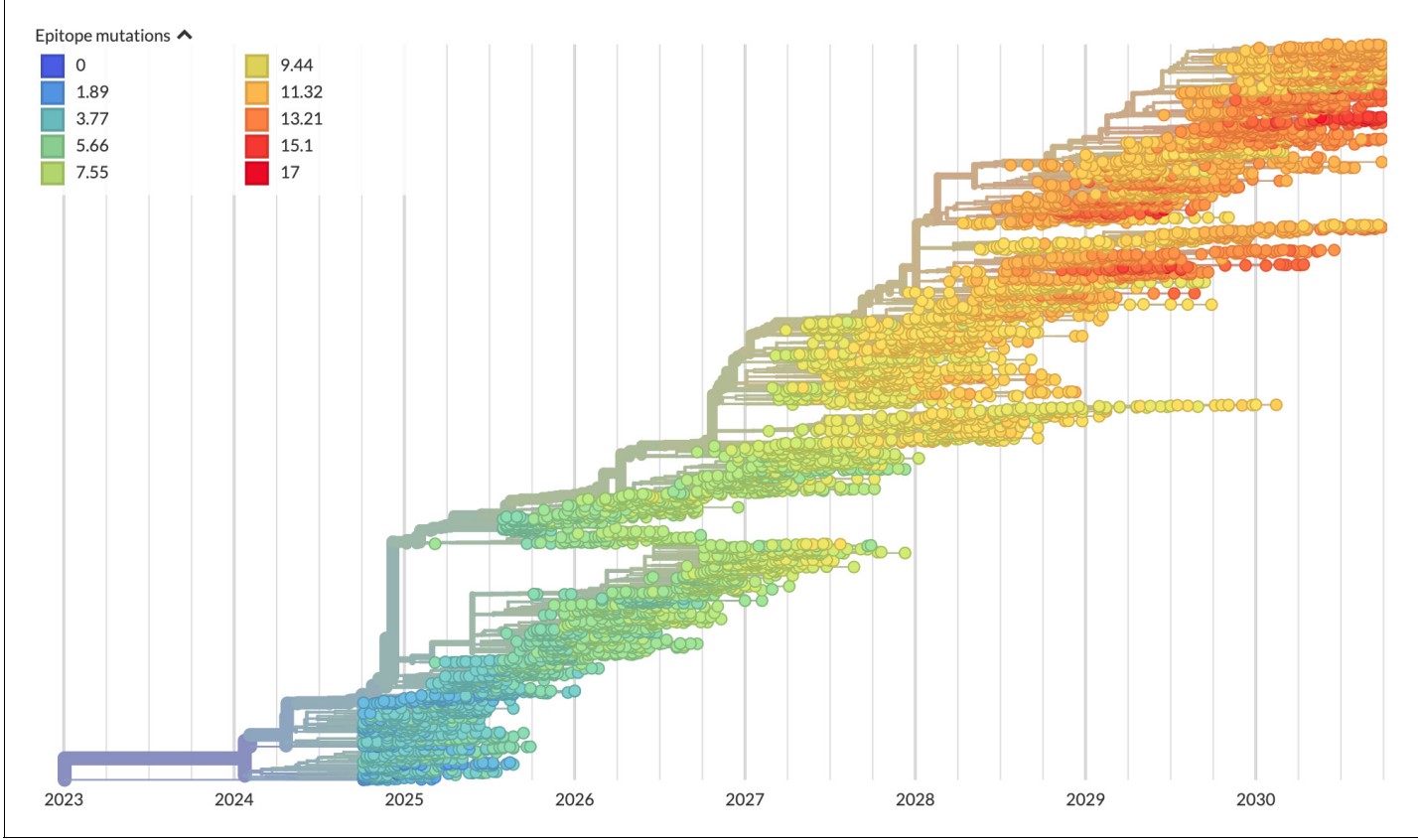

**Figure 3.** Phylogeny of H3N2-like HA sequences sampled between the 24th and 30th years of simulated evolution. The phylogenetic structure and rate of accumulated epitope and non-epitope mutations match patterns observed in phylogenies of natural sequences. Sample dates were annotated as the generation in the simulation divided by 200 and added to 2000, to acquire realistic date ranges that were compatible with our modeling machinery.

*Figure 6—figure supplement 1A*). The model also accurately predicted the growth of 87% of growing clades and the decline of 58% of declining clades. Model forecasts were increasingly more accurate with increasing initial clade frequencies (*Figure 6—figure supplement 1C*). Next, we counted how often the estimated closest strain to the future population at any given timepoint ranked among the observed top closest strains to the future. We calculated the distance of each present strain to the future as the Hamming distance between the given strain's amino acid sequence and each future strain weighted by the future strain's observed or estimated frequency (*Equations 3 and 4*). The estimated closest strain was in the top first percentile of observed closest strains for half of the validation timepoints and in the top 20th percentile for 100% of timepoints (*Figure 6—figure supplement 1B*). Percentile ranks per strain based on their observed and estimated distances to the future correlated strongly across all strains and timepoints (Spearman's $\rho^2 = 0.87$, *Figure 6—figure supplement 1D*). In contrast, the naive model's forecasts of clade frequencies were considerably less

**Table 2.** Number of epitope and non-epitope mutations per branch by trunk or side branch status for simulated populations.

Epitope sites were defined previously described (*Luksza and Lässig, 2014*). Annotation of trunk and side branch was performed as previously described (*Bedford et al., 2015*). Mutations were calculated for the full validation tree for simulated sequences samples between October of years 10 and 40.

| branch type | epitope mutations | non-epitope mutations | epitope-to-non-epitope ratio |
|---|---|---|---|
| side branch | 590 | 1327 | 0.44 |
| trunk | 23 | 12 | 1.92 |

**Table 3.** Number of epitope and non-epitope mutations per branch by trunk or side branch status for natural populations.

Epitope sites were defined previously described (*Luksza and Lässig, 2014*). Annotation of trunk and side branch was performed as previously described (*Bedford et al., 2015*). Mutations were calculated for the full validation tree for natural sequences samples between 1990 and 2015.

| branch type | epitope mutations | non-epitope mutations | epitope-to-non-epitope ratio |
|---|---|---|---|
| side branch | 485 | 1177 | 0.41 |
| trunk | 50 | 32 | 1.56 |

accurate (*Figure 6—figure supplement 2C*). However, the naive model's estimated closest strains to the future were consistently in the top fifth percentile of observed distances to the future and the correlation of its estimated percentile ranks and the observed ranks was strong (Spearman's $\rho^2 = 0.78$, *Figure 6—figure supplement 2B D*). These results suggested that estimating a single closest strain to the future is a more tractable problem than estimating the future frequencies of clades.

Finally, we tested all of our models on out-of-sample data. Specifically, we fixed the coefficients of each model to the average values across the validation period and applied the resulting models to the next 9 years of previously unobserved simulated data. A standard expectation from machine learning is that models will perform worse on test data due to overfitting to training data. Despite this expectation, we found that all models except for the individual epitope mutation models consistently outperformed the naive model across the out-of-sample data (*Figure 4*, *Figure 5*, *Figure 5—figure supplement 1*, *Table 4*). The composite model of mutational load and LBI appeared to

**Table 4.** Simulated population model coefficients and performance on validation and test data ordered from best to worst by distance to the future in the validation analysis.

Coefficients are the mean ± standard deviation for each metric in a given model across 33 training windows. Distance to the future (mean ± standard deviation) measures the distance in amino acids between estimated and observed future populations. Distances annotated with asterisks (*) were significantly closer to the future than the naive model as measured by bootstrap tests (see Methods and *Figure 14*). The number of times (and percentage of total times) each model outperformed the naive model measures the benefit of each model over a model than estimates no change between current and future populations. Test results are based on 18 time-points not observed during model training and validation. Source data are in *Table 4—source data 1* and *2*.

| Model | Coefficients | Distance to future (AAs) | | Model > naive | |
|---|---|---|---|---|---|
| | | Validation | Test | Validation | Test |
| true fitness | 9.37 +/− 0.92 | 6.82 +/− 1.52* | 7.38 +/− 1.89* | 32 (97%) | 16 (89%) |
| LBI | 1.31 +/− 0.33 | 7.24 +/− 1.66* | 7.10 +/− 1.19* | 32 (97%) | 18 (100%) |
| + mutational load | −1.77 +/− 0.49 | | | | |
| LBI | 2.26 +/− 1.06 | 7.57 +/− 1.85* | 7.51 +/− 1.20* | 29 (88%) | 17 (94%) |
| delta frequency | 1.46 +/− 0.44 | 8.13 +/− 1.44* | 8.65 +/− 1.99* | 26 (79%) | 13 (72%) |
| epitope ancestor | 0.35 +/− 0.07 | 8.20 +/− 1.39* | 8.17 +/− 1.52* | 29 (88%) | 17 (94%) |
| + mutational load | −1.57 +/− 0.13 | | | | |
| mutational load | −1.49 +/− 0.12 | 8.27 +/− 1.35* | 8.20 +/− 1.50* | 29 (88%) | 17 (94%) |
| epitope antigenic novelty | 0.03 +/− 0.19 | 8.33 +/− 1.35* | 8.22 +/− 1.51* | 28 (85%) | 17 (94%) |
| + mutational load | −1.38 +/− 0.39 | | | | |
| epitope ancestor | 0.14 +/− 0.11 | 8.96 +/− 1.35 | 9.03 +/− 1.68* | 20 (61%) | 13 (72%) |
| naive | 0.00 +/− 0.00 | 8.97 +/− 1.35 | 9.07 +/− 1.70 | 0 (0%) | 0 (0%) |
| epitope antigenic novelty | −0.03 +/− 0.19 | 9.03 +/− 1.37 | 9.07 +/− 1.69 | 14 (42%) | 7 (39%) |

The online version of this article includes the following source data for Table 4:

Source data 1. Coefficients for models fit to simulated populations.

Source data 2. Distances to the future for models fit to simulated populations.

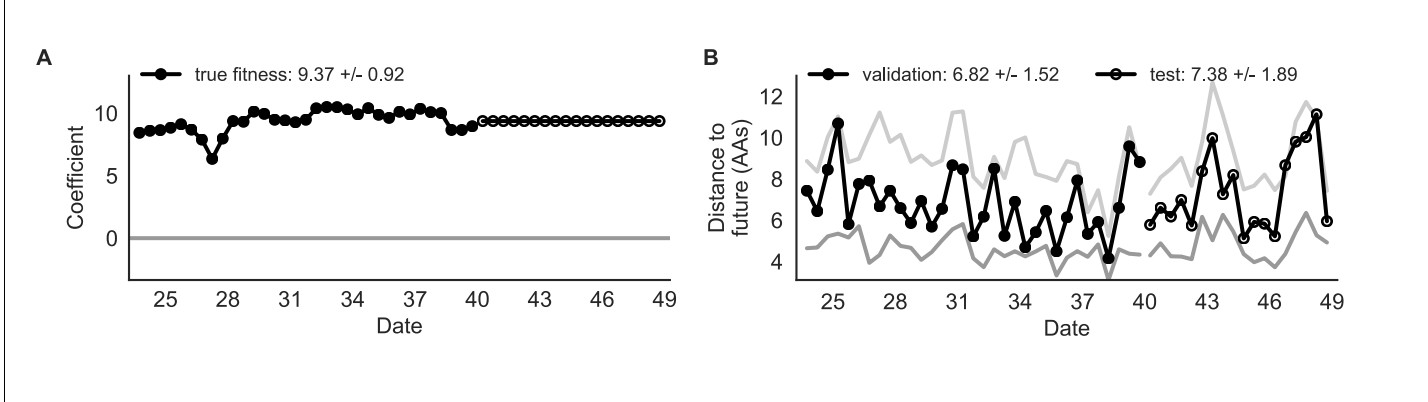

**Figure 4.** Simulated population model coefficients and distances between projected and observed future populations as measured in amino acids (AAs). (**A**) Coefficients are shown per validation timepoint (solid circles, N = 33) with the mean ± standard deviation in the top-left corner. For model testing, coefficients were fixed to their mean values from training/validation and applied to out-of-sample test data (open circles, N = 18). (**B**) Distances between projected and observed populations are shown per validation timepoint (solid black circles) or test timepoint (open black circles). The mean ± standard deviation of distances per validation timepoint are shown in the top-left of each panel. Corresponding values per test timepoint are in the top-right. The naive model's distances to the future for validation and test timepoints (light gray) were 8.97 ± 1.35 AAs and 9.07 ± 1.70 AAs, respectively. The corresponding lower bounds on the estimated distance to the future (dark gray) were 4.57 ± 0.61 AAs and 4.85 ± 0.82 AAs. Source data are in *Table 4—source data 1* and *2*.

outperform the true fitness metric with average distance to the future of 7.10 ± 1.19 compared to 7.38 ± 1.89, respectively. However, we did not find a significant difference between these models by bootstrap testing (*Table 5*) and could not rule out fluctuations in model performance across a relatively small number of data points.

As with our validation dataset, we tested the true fitness model's ability to recapitulate clade dynamics and select optimal individual strains from the test data. While observed and estimated clade frequency fold changes correlated more weakly for test data (Pearson's $R^2 = 0.14$), the accuracies of clade growth and decline predictions remained similar at 82% and 53%, respectively (*Figure 6A*). We observed higher absolute forecast errors in the test data with higher errors for clades between 40% and 60% initial frequencies (*Figure 6C*). The estimated closest strain was higher than the top first percentile of observed closest strains for half of the test timepoints and in the top 20th percentile for 16 (89%) of 18 of timepoints (*Figure 6B*). Observed and estimated strain ranks remained strongly correlated across all strains and timepoints (Spearman's $\rho^2 = 0.80$, *Figure 6D*). The naive model performed comparatively well on these test data with all its estimated closest strains to the future in the top 20th percentile and a slightly higher correlation between observed and estimated percentile ranks than the true fitness model (Spearman's $\rho^2 = 0.82$, *Figure 6—figure supplement 3*). These results confirmed that our approach of minimizing the distance between yearly populations could simultaneously capture clade-level dynamics of simulated influenza populations and identify individual strains that are most representative of future populations. However, they also supported the earlier finding that clade frequency forecasts may be inherently more challenging than identification of the closest strain to the future.

## Models reflect historical patterns of H3N2 evolution

Next, we trained and validated models for individual fitness predictors using 25 years of natural H3N2 populations spanning from October 1, 1990 to October 1, 2015. We held out strains collected after October 1, 2015 up through October 1, 2019 for model testing (*Figure 7*). In addition to the sequence-only models we tested on simulated populations, we also fit models for our new fitness metrics based on experimental phenotypes including HI antigenic novelty and DMS mutational effects. We hypothesized that both HI and DMS metrics would be assigned positive coefficients, as they estimate increased antigenic drift and beneficial mutations, respectively. As antigenic drift is generally considered to be the primary evolutionary pressure on natural H3N2 populations (*Smith et al., 2004*; *Bedford et al., 2014*; *Luksza and Lässig, 2014*), we expected that epitope and HI antigenic novelty would be individually more predictive than mutational load or DMS mutational

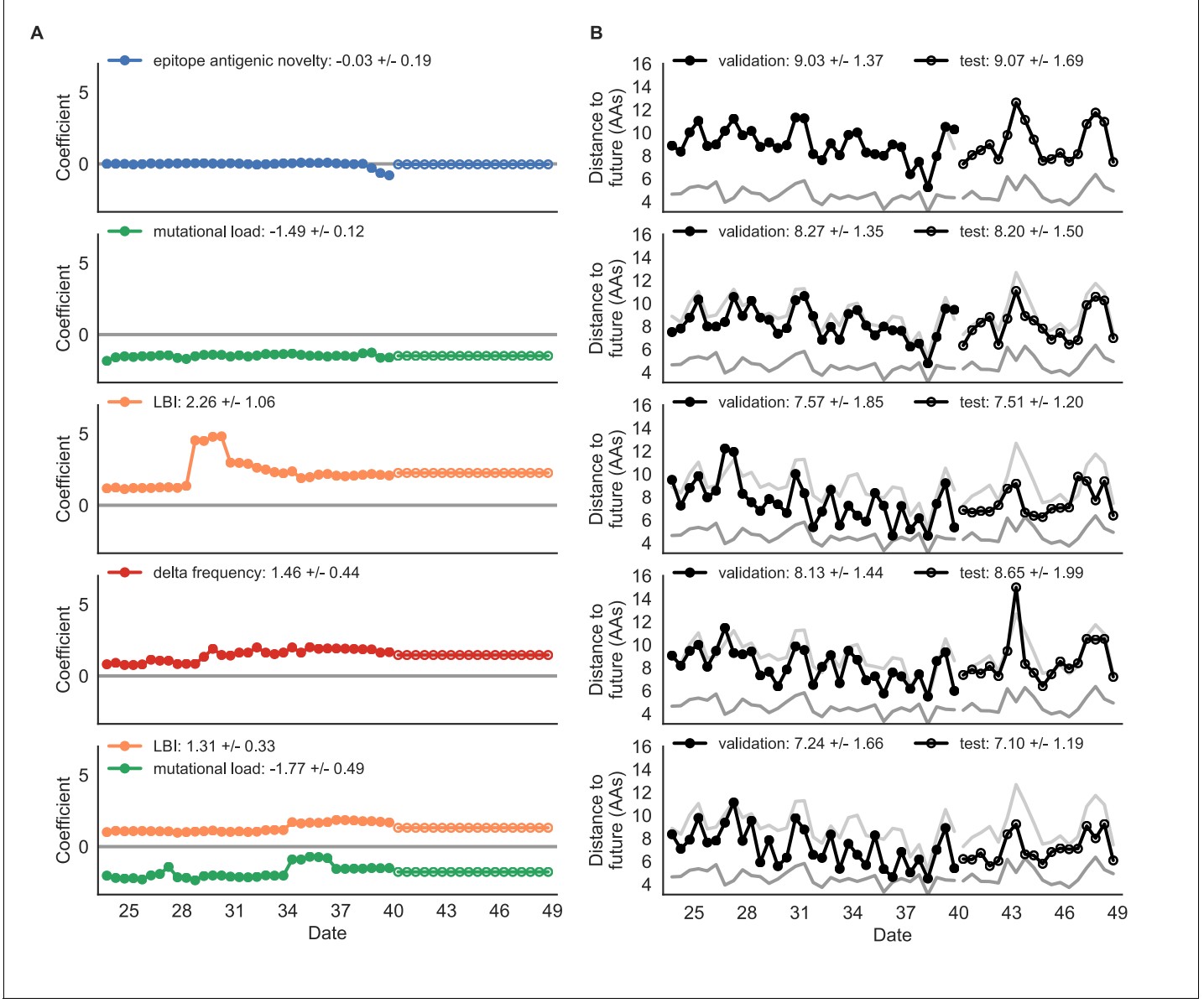

**Figure 5.** Simulated population model coefficients and distances to the future for individual biologically-informed fitness metrics and the best composite model. (**A**) Coefficients and (**B**) distances are shown per validation and test timepoint as in *Figure 4*. Source data are in *Table 4—source data 1* and *2*.

The online version of this article includes the following figure supplement(s) for figure 5:

**Figure supplement 1.** Composite model coefficients and distances to the future for models fit to simulated populations.

effects. Previous research (*Neher et al., 2014*) and our simulation results also led us to expect that LBI and delta frequency would outperform other individual mechanistic metrics. As the earliest measurements from focus reduction assays (FRAs) date back to 2012, we could not train, validate, and test FRA antigenic novelty models in parallel with the HI antigenic novelty models.

Biologically-informed metrics generally performed better than the naive model with the exceptions of the epitope antigenic novelty and DMS mutational effects (*Figure 8* and *Table 6*). The naive model estimated an average distance between natural H3N2 populations of 6.40 ± 1.36 AAs. The lower bound for how well any model could perform, 2.60 ± 0.89 AAs, was considerably lower than the corresponding bounds for simulated populations. The average improvement of the sequence-only models over the naive model was consistently lower than the same models in simulated

**Table 5.** Comparison of composite and individual model distances to the future by bootstrap test (see Materials and methods). The effect size of differences between models in amino acids is given by the mean and standard deviation of the bootstrap distributions. The p values represent the proportion of n = 10,000 bootstrap samples where the mean difference was greater than or equal to zero.

| sample | error type | individual model | composite model | bootstrap mean | bootstrap std | p value |
|---|---|---|---|---|---|---|
| simulated | validation | true fitness | mutational load + LBI | 0.42 | 0.23 | 0.9644 |
| simulated | validation | mutational load | mutational load + LBI | −1.03 | 0.21 | <0.0001 |
| simulated | validation | LBI | mutational load + LBI | −0.33 | 0.14 | 0.0091 |
| simulated | test | true fitness | mutational load + LBI | −0.28 | 0.26 | 0.1392 |
| simulated | test | mutational load | mutational load + LBI | −1.11 | 0.25 | <0.0001 |
| simulated | test | LBI | mutational load + LBI | −0.42 | 0.16 | 0.0001 |
| natural | validation | mutational load | mutational load + LBI | −0.69 | 0.28 | 0.0036 |
| natural | validation | LBI | mutational load + LBI | −0.23 | 0.09 | 0.0025 |
| natural | validation | mutational load | mutational load + HI antigenic novelty | −0.31 | 0.18 | 0.0417 |
| natural | validation | HI antigenic novelty | mutational load + HI antigenic novelty | −0.18 | 0.11 | 0.0513 |
| natural | test | mutational load | mutational load + LBI | 1.19 | 0.79 | 0.9432 |
| natural | test | LBI | mutational load + LBI | −0.70 | 0.24 | <0.0001 |
| natural | test | mutational load | mutational load + HI antigenic novelty | −0.56 | 0.33 | 0.0133 |
| natural | test | HI antigenic novelty | mutational load + HI antigenic novelty | −0.24 | 0.18 | 0.0999 |

populations. This reduced performance may have been caused by both the relatively reduced diversity between years in natural populations and the fact that our simple models do not capture all drivers of evolution in natural H3N2 populations.

Of the two metrics for antigenic drift, HI antigenic novelty consistently outperformed epitope antigenic novelty (*Table 6*). HI antigenic novelty estimated an average distance to the future of 6.01 ± 1.50 AAs and outperformed the naive model at 16 of 23 timepoints (70%). The coefficient for HI antigenic novelty remained stable across all timepoints (*Figure 8*). In contrast, epitope antigenic novelty estimated a distance of 7.13 ± 1.47 AAs and only outperformed the naive model at seven timepoints (30%). Epitope antigenic novelty was also the only metric whose coefficient started at a positive value (1.17 ± 0.03 on average prior to October 2009) and transitioned to a negative value through the validation period (−0.19 ± 0.34 on average for October 2009 and after). This strong coefficient for the first half of training windows indicated that, unlike the results for simulated populations, the nonlinear antigenic novelty metric was historically an effective measure of antigenic drift. The historical importance of the epitope sites used for this metric was further supported by the relative enrichment of mutations at these sites for the most successful 'trunk' lineages of natural populations compared to side branch lineages (*Table 3*).

These results led us to hypothesize that the contribution of these specific epitope sites to antigenic drift has weakened over time. Importantly, these 49 epitope sites were originally selected by *Luksza and Lässig, 2014* from a previous historical survey of sites with beneficial mutations between 1968–2005 (*Shih et al., 2007*). If the beneficial effects of mutations at these sites were due to historical contingency rather than a constant contribution to antigenic drift, we would expect models based on these sites to perform well until 2005 and then overfit relative to future data. Indeed, the epitope antigenic novelty model outperforms the naive model for the first three validation timepoints until it has to predict to April 2006. To test this hypothesis, we identified a new set of beneficial sites across our entire validation period of October 1990 through October 2015. Inspired by the original approach of *Shih et al., 2007*, we identified 25 sites in HA1 where mutations rapidly swept through the global population, including 12 that were also present in the original set of 49 sites. We fit an antigenic novelty model to these 25 sites across the complete validation period and dubbed this the 'oracle antigenic novelty' model, as it benefited from knowledge of the future in its forecasts. The oracle model produced a consistently positive coefficient across all training windows (0.80 ± 0.21) and consistently outperformed the original epitope model with an average distance to

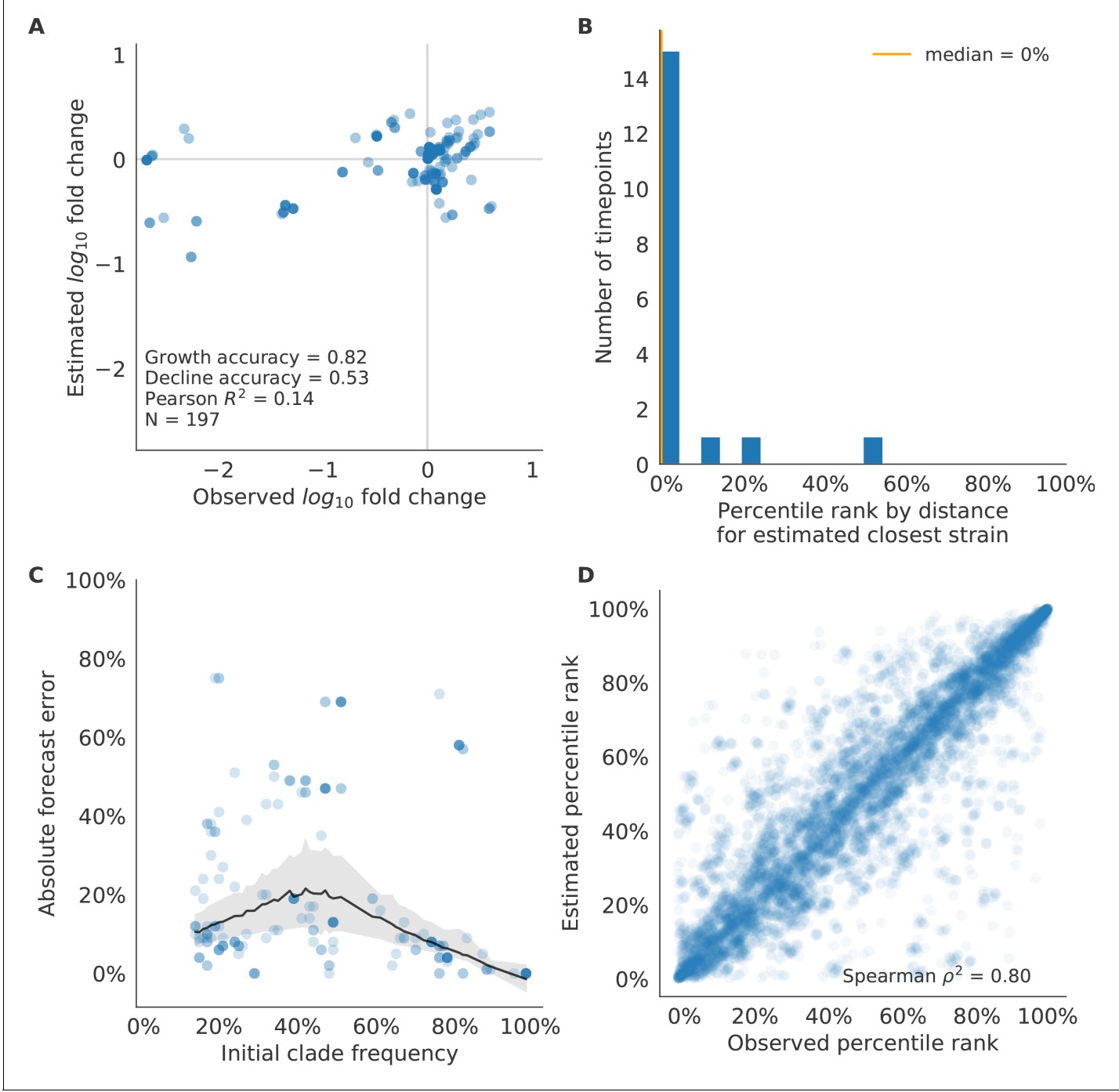

**Figure 6.** Test of best model for simulated populations (true fitness) using 9 years (18 timepoints) of previously unobserved test data and fixed model coefficients. These timepoints correspond to the open circles in *Figure 2*. (A) The correlation of log estimated clade frequency fold change, $\log_{10} \frac{\hat{x}(t+\Delta t)}{x(t)}$, and log observed clade frequency fold change, $\log_{10} \frac{x(t+\Delta t)}{x(t)}$, shows the model's ability to capture clade-level dynamics without explicitly optimizing for clade frequency targets. (B) The rank of the estimated closest strain based on its distance to the future in the best model was in the top 20th percentile for 89% of 18 timepoints, confirming that the model makes a good choice when forced to select a single representative strain for the future population. (C) Absolute forecast error for clades shown in A by their initial frequency with a mean LOESS fit (solid black line) and 95% confidence intervals (gray shading) based on 100 bootstraps. (D) The correlation of all strains at all timepoints by the percentile rank of their observed and estimated distances to the future. The corresponding results for the naive model are shown in *Figure 6—figure supplement 3*.

The online version of this article includes the following figure supplement(s) for figure 6:

**Figure supplement 1.** Validation of best model for simulated populations of H3N2-like viruses.

*Figure 6 continued on next page*

*Figure 6 continued*

**Figure supplement 2.** Validation of naive model for simulated populations of H3N2-like viruses.
**Figure supplement 3.** Test of naive model for simulated populations of H3N2-like viruses.

the future of 5.71 ± 1.27 AAs (*Figure 8—figure supplement 1*). These results support our hypothesis that the fitness benefit of mutations at the original 49 sites was due to historical contingency and that the success of previous epitope models based on these sites was partly due to 'borrowing from the future'. We suspect that our HI antigenic novelty model benefits from its ability to constantly update its antigenic model at each timepoint with recent experimental phenotypes, while the epitope antigenic novelty metric is forced to give a constant weight to the same 49 sites throughout time.

Of the two metrics for functional constraint, mutational load outperformed DMS mutational effects, with an average distance to the future of 6.14 ± 1.37 AAs compared to 6.75 ± 1.95 AAs, respectively. In contrast to the original *Luksza and Lässig, 2014* model, where the coefficient of the mutational load metric was fixed at −0.5, our model learned a consistently stronger coefficient of −0.99 ± 0.30. Notably, the best performance of the DMS mutational effects model was forecasting from April 2007 to April 2008 when the major clade containing A/Perth/16/2009 was first emerging. This result is consistent with the DMS model overfitting to the evolutionary history of the background strain used to perform the DMS experiments. Alternate implementations of less background-dependent DMS metrics never performed better than the mutational load metric (*Table 6*, Materials and methods). Thus, we find that a simple model where any mutation at non-epitope sites is deleterious is more predictive of global viral success than a more comprehensive biophysical model based on measured mutational effects of a single strain.

LBI was the best individual metric by average distance to the future (*Figure 8*) and tied mutational load by outperforming the naive model at 17 (74%) timepoints (*Table 6*). Delta frequency performed worse than LBI and HI antigenic novelty and was comparable to mutational load. While delta frequency should, in principle, measure the same aspect of viral fitness as LBI, these results show that the current implementations of these metrics represent qualitatively different fitness components. The LBI and mutational load might also be predictive for reasons other than correlation with fitness, see Discussion.

To test whether composite models could outperform individual fitness metrics for natural populations, we fit models based on combinations of best individual metrics representing antigenic drift, functional constraint, and clade growth. Specifically, we fit models based on HI antigenic novelty and mutational load, mutational load and LBI, and all three of these metrics together. We anticipated that if these metrics all represented distinct, mutually beneficial components of viral fitness, these composite models should perform better than individual models with consistent coefficients for each metric.

Both two-metric composite models modestly outperformed their corresponding individual models (*Table 6*, *Figure 9*, and *Table 5*). The composite of mutational load and LBI performed the best overall with an average distance to the future of 5.44 ± 1.80 AAs. The relative stability of the coefficients for the metrics in the two-metric models suggested that these metrics represented complementary components of viral fitness. In contrast, the three-metric model strongly preferred the HI antigenic novelty and mutational load metrics over LBI for the entire validation period, producing an average LBI coefficient of −0.04 ± 0.09. Overall, the gain by combining multiple predictors was limited and the sensitivity of coefficients to the set of metrics included in the model suggests that there is substantial overlap in predictive value of different metrics.

As with the simulated populations, we validated the performance of the best model for natural populations using estimated and observed clade frequency fold changes and the ranking of estimated closest strains compared to the observed closest strains to future populations. The composite model of mutational load and LBI effectively captured clade dynamics with a fold change correlation of $R^2 = 0.35$ and growth and decline accuracies of 87% and 89%, respectively (*Figure 10—figure supplement 1A*). Absolute forecasting error declined noticeably for clades with initial frequencies above 60%, but generally this error remained below 20% on average (*Figure 10—figure supplement 1C*). The estimated closest strain from this model was in the top first percentile of observed

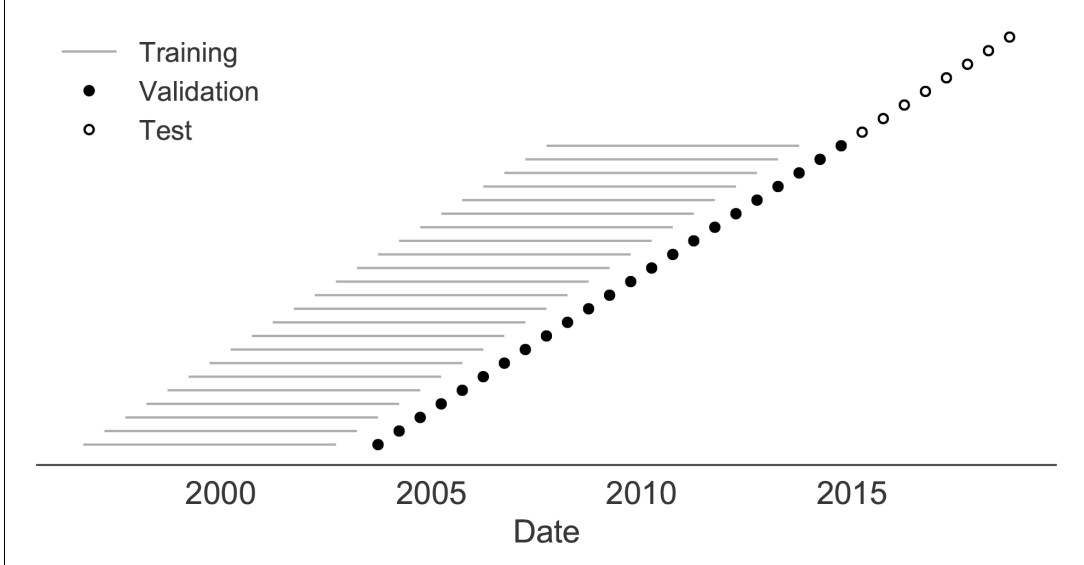

**Figure 7.** Time-series cross-validation scheme for natural populations. Models were trained in six-year sliding windows (gray lines) and validated on out-of-sample data from validation timepoints (filled circles). Validation results from 25 years of data were used to iteratively tune model hyperparameters. After fixing hyperparameters, model coefficients were fixed at the mean values across all training windows. Fixed coefficients were applied to four years of new out-of-sample test data (open circles) to estimate true forecast errors.

closest strains for half of the validation timepoints and in the top 20th percentile for 20 (87%) of 23 timepoints (*Figure 10—figure supplement 1B*). This pattern held across all strains and timepoints with a strong correlation between observed and estimated strain ranks (Spearman's $\rho^2 = 0.66$, *Figure 10—figure supplement 1D*). The naive model's performance repeated the pattern we observed with simulated populations: it made poor forecasts of absolute clade frequencies, but its estimated closest strains to the future were consistently highly ranked among the observed closest strains (*Figure 10—figure supplement 2B C*).

Finally, we tested the performance of all models on out-of-sample data collected from October 1, 2015 through October 1, 2019. We anticipated that most models would perform worse on truly out-of-sample data than on validation data. Correspondingly, only the three models with the HI antigenic novelty metric significantly outperformed the naive model on the test data (*Table 6*). The composite of HI antigenic novelty and mutational load performed modestly, although not significantly, better than the individual HI antigenic novelty model (*Table 5*). Surprisingly, the best model for the validation data – mutational load and LBI – was one of the worst models for the test data with an average distance to the future of 7.70 ± 3.53 AAs. The individual LBI model was the worst model, while mutational load continued to perform well with test data. LBI performed especially poorly in the last two test timepoints of April and October 2018 (*Figure 8*). These timepoints correspond to the dominance and sudden decline of a reassortant clade named A2/re (*Potter et al., 2019*). By April 2018, the A2/re clade had risen to a global frequency over 50% from less than 15% the previous year, despite an absence of antigenic drift. By October 2018, this clade had declined in frequency to approximately 30% and, by October 2019, it had gone extinct. That LBI incorrectly predicted the success of this reassortant clade highlights a major limitation of growth-based fitness metrics and a corresponding benefit of more mechanistic metrics that explicitly measure antigenic drift and functional constraint. However, we cannot rule out the alternate possibility that the LBI model was overfit to the training data.

After identifying the composite HI antigenic novelty and mutational load model as the best model on out-of-sample data, we tested this model's ability to detect clade dynamics and select individual closest strains to the future for vaccine composition. The composite model partially captured clade dynamics with a Pearson's correlation of $R^2 = 0.46$ between observed and estimated growth ratios and growth and decline accuracies of 52% and 58%, respectively (*Figure 10A*). The mean absolute forecasting error with this model was consistently less than 20%, regardless of the initial clade

frequency (*Figure 10C*). The estimated closest strain from this model was in the top first percentile of observed closest strains for half of the validation timepoints and in the top 20th percentile for 100% of timepoints (*Figure 10B*). Similarly, the observed and estimated strain ranks strongly correlated (Spearman's $\rho^2 = 0.72$) across all strains and test timepoints (*Figure 10D*). The estimated strain ranks of the naive model were not as well correlated (Spearman's $\rho^2 = 0.56$), but seven of its eight estimates for the closest strain to the future (88%) were in the top fifth percentile of observed closest strains (*Figure 10—figure supplement 3B D*).

We further evaluated our models' ability to estimate the closest strain to the next season's H3N2 population by comparing our best models' selections to the WHO's vaccine strain selection. For

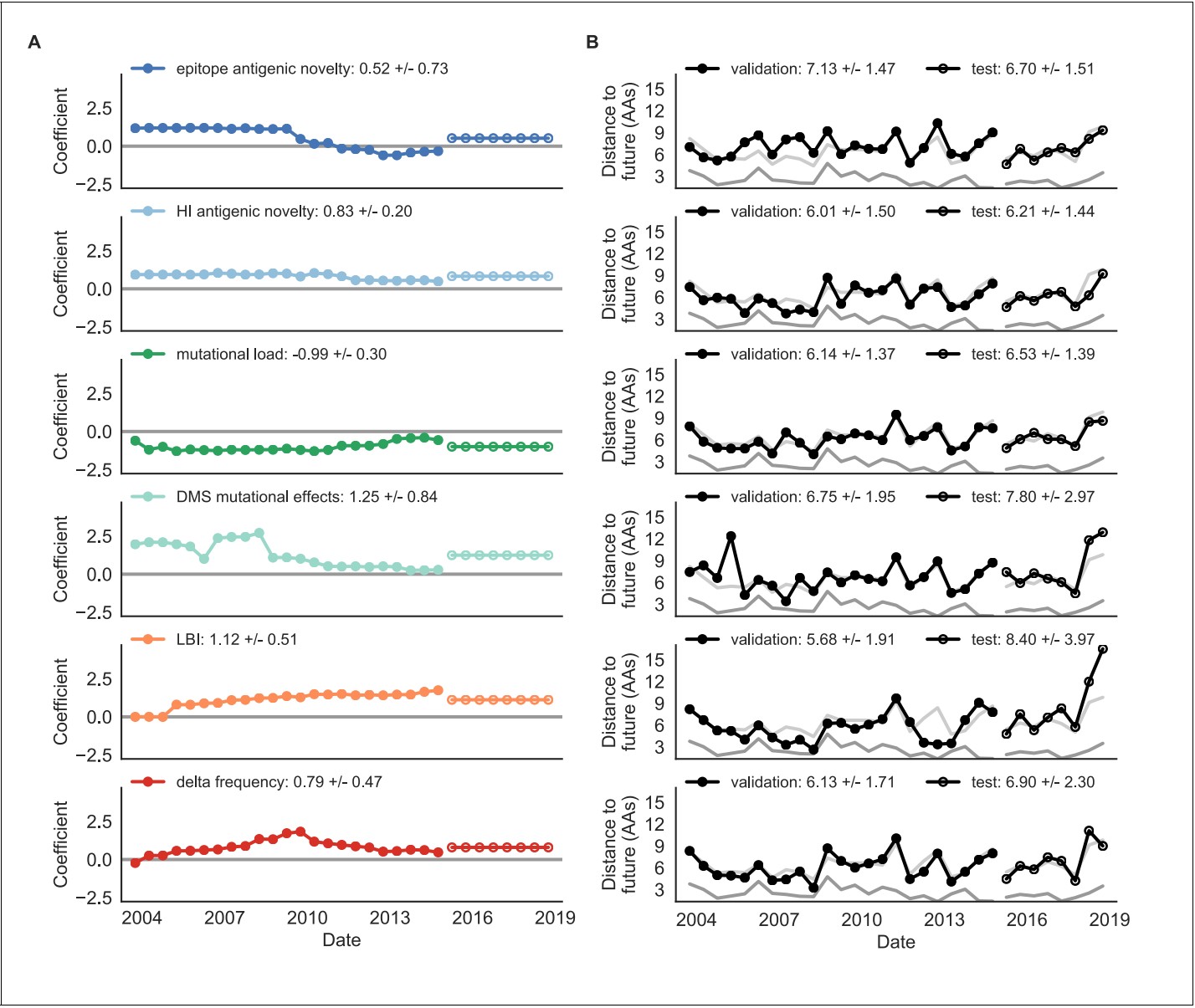

**Figure 8.** Natural population model coefficients and distances to the future for individual biologically-informed fitness metrics. (**A**) Coefficients and (**B**) distances are shown per validation timepoint (N = 23) and test timepoint (N = 8) as in *Figure 4*. The naive model's distance to the future (light gray) was 6.40 ± 1.36 AAs for validation timepoints and 6.82 ± 1.74 AAs for test timepoints. The corresponding lower bounds on the estimated distance to the future (dark gray) were 2.60 ± 0.89 AAs and 2.28 ± 0.61 AAs. Source data are in *Table 6—source data 1* and *2*.

The online version of this article includes the following figure supplement(s) for figure 8:

**Figure supplement 1.** Comparison of epitope-based models with knowledge of the future.

**Table 6.** All model coefficients and performance on validation and test data for natural populations ordered from best to worst by distance to the future, as in **Table 4**.

Distances annotated with asterisks (*) were significantly closer to the future than the naive model as measured by bootstrap tests (see Materials and methods and **Figure 15**). Distances annotated with carets (∧) were not tested for significance relative to the naive model. Validation results are based on 23 timepoints. Test results are based on eight timepoints not observed during model training and validation. Model results for additional variants of fitness metrics including those based on epitope mutations and DMS preferences are included for reference. Source data are in **Table 6—source data 1** and **2**.

| Model | Coefficients | Distance to future (AAs) | | Model > naive | |
| --- | --- | --- | --- | --- | --- |
| | | Validation | Test | Validation | Test |
| mutational load | −0.68 +/− 0.34 | 5.44 +/− 1.80* | 7.70 +/− 3.53 | 18 (78%) | 4 (50%) |
| + LBI | 1.03 +/− 0.40 | | | | |
| LBI | 1.12 +/− 0.51 | 5.68 +/− 1.91* | 8.40 +/− 3.97 | 17 (74%) | 2 (25%) |
| oracle antigenic novelty | 0.80 +/− 0.21 | 5.71 +/− 1.27^ | 8.06 +/− 2.49^ | 18 (78%) | 2 (25%) |
| HI antigenic novelty | 0.89 +/− 0.23 | 5.82 +/− 1.50* | 5.97 +/− 1.47* | 17 (74%) | 6 (75%) |
| + mutational load | −1.01 +/− 0.42 | | | | |
| HI antigenic novelty | 0.90 +/− 0.23 | 5.84 +/− 1.51* | 5.99 +/− 1.46* | 16 (70%) | 6 (75%) |
| + mutational load | −1.00 +/− 0.44 | | | | |
| + LBI | −0.04 +/− 0.09 | | | | |
| HI antigenic novelty | 0.83 +/− 0.20 | 6.01 +/− 1.50* | 6.21 +/− 1.44* | 16 (70%) | 7 (88%) |
| delta frequency | 0.79 +/− 0.47 | 6.13 +/− 1.71* | 6.90 +/− 2.30 | 16 (70%) | 5 (62%) |
| mutational load | −0.99 +/− 0.30 | 6.14 +/− 1.37* | 6.53 +/− 1.39 | 17 (74%) | 6 (75%) |
| Koel epitope antigenic novelty | 0.28 +/− 0.36 | 6.22 +/− 1.26^ | 6.72 +/− 1.51^ | 18 (78%) | 4 (50%) |
| naive | 0.00 +/− 0.00 | 6.40 +/− 1.36 | 6.82 +/− 1.74 | 0 (0%) | 0 (0%) |
| DMS entropy | −0.03 +/− 0.10 | 6.40 +/− 1.36^ | 6.81 +/− 1.73^ | 9 (39%) | 6 (75%) |
| DMS mutational load | −0.02 +/− 0.13 | 6.45 +/− 1.42^ | 6.82 +/− 1.73^ | 7 (30%) | 5 (62%) |
| epitope ancestor | 0.53 +/− 0.52 | 6.60 +/− 1.34 | 6.53 +/− 1.51 | 12 (52%) | 4 (50%) |
| + mutational load | −0.77 +/− 0.32 | | | | |
| DMS mutational effects | 1.25 +/− 0.84 | 6.75 +/− 1.95 | 7.80 +/− 2.97 | 11 (48%) | 4 (50%) |
| Wolf epitope antigenic novelty | 0.31 +/− 0.51 | 6.83 +/− 1.30^ | 6.97 +/− 1.41^ | 4 (17%) | 3 (38%) |
| epitope ancestor | 0.23 +/− 0.51 | 6.89 +/− 1.39^ | 6.82 +/− 1.67^ | 8 (35%) | 4 (50%) |
| epitope antigenic novelty | 0.57 +/− 0.77 | 6.89 +/− 1.42 | 6.46 +/− 1.31 | 7 (30%) | 4 (50%) |
| + mutational load | −0.77 +/− 0.27 | | | | |
| epitope antigenic novelty | 0.52 +/− 0.73 | 7.13 +/− 1.47 | 6.70 +/− 1.51 | 7 (30%) | 5 (62%) |

The online version of this article includes the following source data for Table 6:

Source data 1. Coefficients for models fit to natural populations.

Source data 2. Distances to the future for models fit to natural populations.Clustering of amino acid sequences for visualization.

each season when the WHO selected a new vaccine strain and one year of future data existed in our validation or test periods, we measured the observed distance of that strain's sequence to the future and the corresponding distances to the future for the observed closest strains (*Equation 3*). We compared these distances to those of the closest strains to the future as estimated by our best models for the validation period (mutational load and LBI) and the test period (HI antigenic novelty and mutational load) using *Equation 4*. The observed closest strain to the future represents the centroid of the observed future population, while the estimated closest strains are the models' predictions of that future population's centroid. The mutational load and LBI model selected strains that were as close or closer to the future than the corresponding vaccine strain for 10 (83%) of the 12 seasons with vaccine updates (*Figure 11*). On average, the strains selected by this model were closer to future than the vaccine strain by 1.93 AAs (*Figure 11—figure supplement 1*). For the two seasons that the model selected more distant strains than the vaccine strain, the mean distance relative to

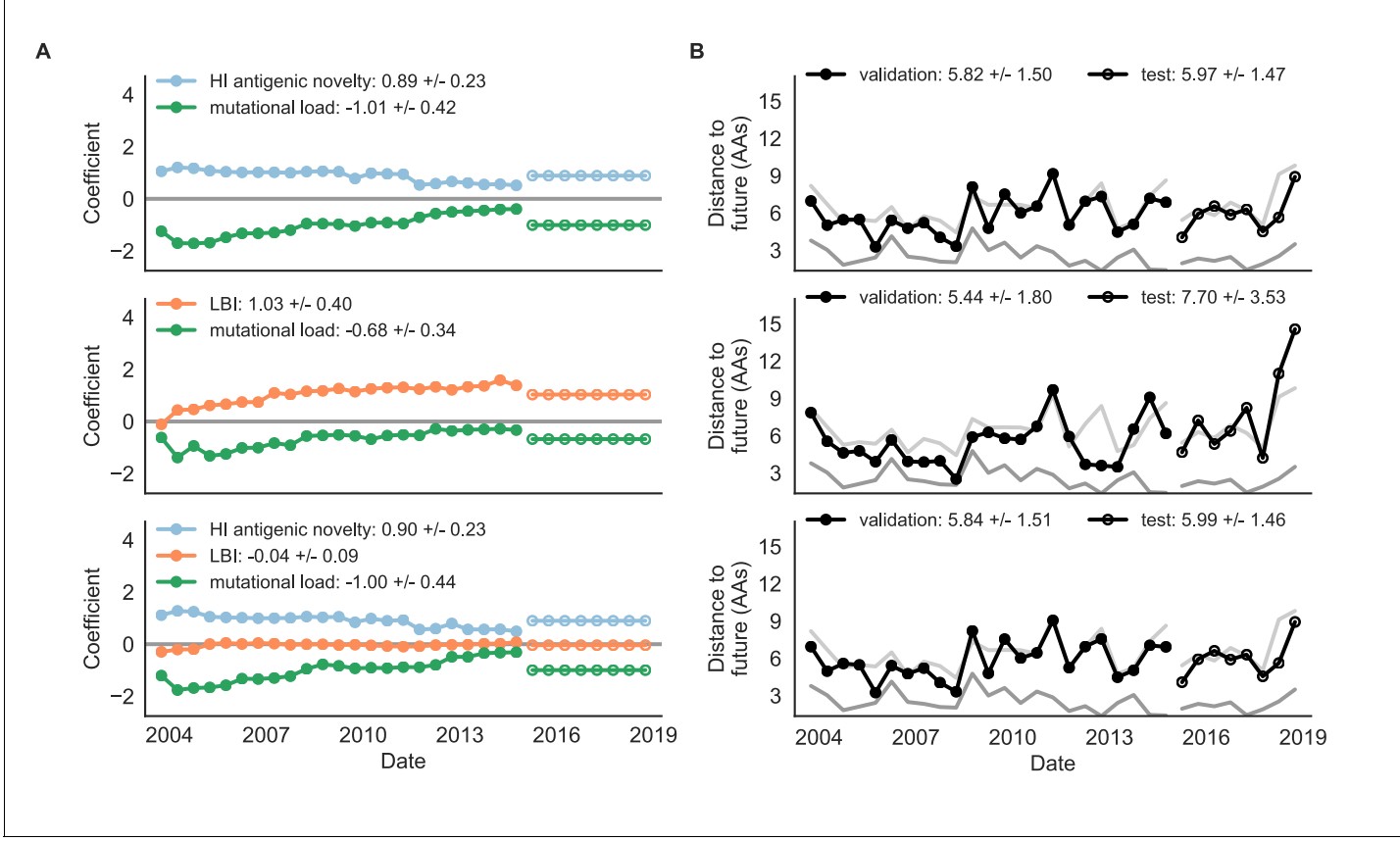

**Figure 9.** Natural population model coefficients and distances to the future for composite fitness metrics. (**A**) Coefficients and (**B**) distances are shown per validation timepoint (N = 23) and test timepoint (N = 8) as in *Figure 4*. Source data are in *Table 6—source data 1* and *2*.

The online version of this article includes the following figure supplement(s) for figure 9:

**Figure supplement 1.** Composite models fit to most recent data from natural populations.

the vaccine strain was 1.58 AAs. The HI antigenic novelty and mutational load model performed similarly by identifying strains as close or closer to the future for 11 (92%) seasons with an average improvement over the vaccine strains of 2.33 AAs. For the one season that the model selected a more distant strain, that selected strain was 0.75 AAs farther from the future than the vaccine strain. Interestingly, the strains selected by the naive model were always better than the selected vaccine strain. Since the naive model predicts that the future will be identical to the present, these strains represent the centroid of each current population. With an average improvement over the vaccine strains of 2.19 AAs, the naive model performed consistently better than the LBI-based model and nearly as well as the HI-based model. These results were consistent with our earlier observations that the naive model often performs as well as biologically-informed models when estimating a single closest strain to the future.

## Historically-trained models enable real-time, actionable forecasts

To enable real-time forecasts, we integrated our forecasting framework into our existing open source pathogen surveillance application, Nextstrain (*Hadfield et al., 2018*). Prior to finalizing our model coefficients for use in Nextstrain, we tested whether our three best composite models could be improved by learning new coefficients per timepoint from the test data. Additionally, we evaluated a composite of FRA antigenic novelty and mutational load. Since the earliest FRA data were from 2012, we anticipated that there were enough measurements to fit a model across the test data time interval. If modern H3N2 strains continue to perform poorly in HI assays, the FRA-based assay will be critical for future forecasting efforts.

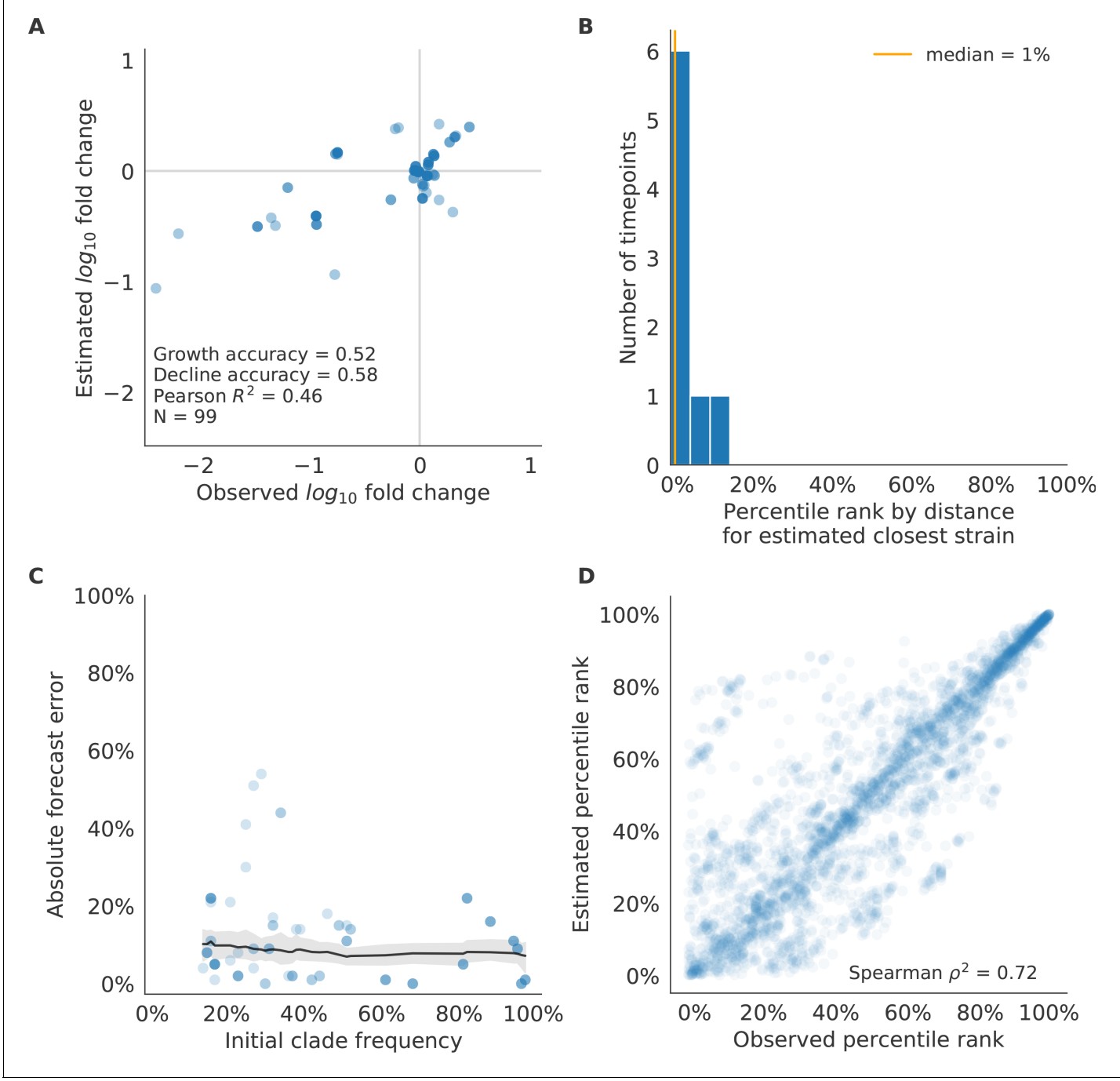

**Figure 10.** Test of best model for natural populations of H3N2 viruses, the composite model of HI antigenic novelty and mutational load, across eight timepoints. These timepoints correspond to the open circles in *Figure 7*. (**A**) The correlation of estimated and observed clade frequency fold changes shows the model's ability to capture clade-level dynamics without explicitly optimizing for clade frequency targets. (**B**) The rank of the estimated closest strain based on its distance to the future for eight timepoints. The estimated closest strain was in the top 20th percentile of observed closest strains for 100% of timepoints. (**C**) Absolute forecast error for clades shown in A by their initial frequency with a mean LOESS fit (solid black line) and 95% confidence intervals (gray shading) based on 100 bootstraps. (**D**) The correlation of all strains at all timepoints by the percentile rank of their observed and estimated distances to the future. The corresponding results for the naive model are shown in *Figure 10—figure supplement 3*.

The online version of this article includes the following figure supplement(s) for figure 10:

**Figure supplement 1.** Validation of best model for natural populations of H3N2 viruses.

**Figure supplement 2.** Validation of naive model for natural populations of H3N2 viruses.

**Figure supplement 3.** Test of naive model for natural populations of H3N2 viruses.

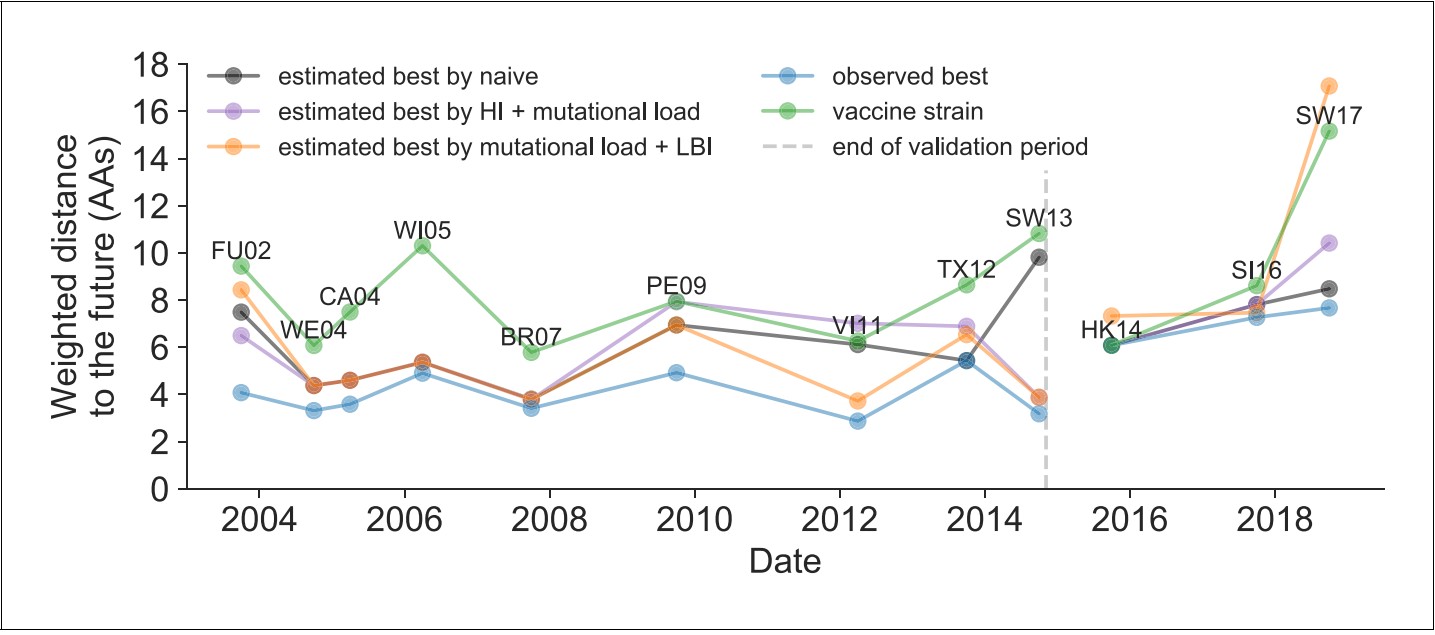

**Figure 11.** Observed distance to natural H3N2 populations one year into the future for each vaccine strain (green) and the observed (blue) and estimated closest strains to the future by the mutational load and LBI model (orange), the HI antigenic novelty and mutational load model (purple), and the naive model (black). Vaccine strains were assigned to the validation or test timepoint closest to the date they were selected by the WHO. The weighted distance to the future for each strain was calculated from their amino acid sequences and the frequencies and sequences of the corresponding population one year in the future. Vaccine strain names are abbreviated from A/Fujian/411/2002, A/Wellington/1/2004, A/California/7/2004, A/Wisconsin/67/2005, A/Brisbane/10/2007, A/Perth/16/2009, A/Victoria/361/2011, A/Texas/50/2012, A/Switzerland/9715293/2013, A/HongKong/4801/2014, A/Singapore/Infimh-16-0019/2016, and A/Switzerland/8060/2017. Source data are available in *Figure 11—source data 1*.

The online version of this article includes the following source data and figure supplement(s) for figure 11:

**Source data 1.** Weighted distances to the future per strain by strain type and timepoint.

**Figure supplement 1.** Relative improvement of model selections over vaccine strains.

Two of three models performed worse after refitting coefficients to the test data than their original fixed coefficient implementations (*Figure 9—figure supplement 1*). While, the mutational load and LBI model improved considerably over its original performance, it still performed worse than the naive model on average. These results confirmed that the coefficients for our selected best model would be most accurate for live forecasts. Interestingly, the FRA antigenic novelty metric received a consistently positive coefficient of $1.40 \pm 0.24$ in its composite with mutational load. Unfortunately, this model performed considerably worse than the corresponding HI-based model. These results suggest that we may need more FRA data across a longer historical timespan to train a model that could replace the HI-based model.

After confirming the coefficients for our best model of HI antigenic novelty and mutational load, we inspected forecasts of H3N2 clades using all data available up through June 6, 2020. Consistent with an average two-month lag between data collection and submission, the most recent data were collected up to April 1, 2020 and made our forecasts from this timepoint to April 1, 2021. Of the five major currently circulating clades, our model predicted growth of the clades 3c3.A and A1b/94N and decline of clades A1b/135K, A1b/137F, and A1b/197R (*Figure 12*). To aid with identification of potential vaccine candidates for the next season, we annotated strains in the phylogeny by their estimated distance to the future based on our best model (*Figure 13*).

## Discussion

We have developed and rigorously tested a novel, open source framework for forecasting the long-term evolution of seasonal influenza H3N2 by estimating the sequence composition of future populations. A key innovation of this framework is its ability to directly compare viral populations between seasons using the earth mover's distance metric (*Rubner et al., 1998*) and eliminate unavoidably

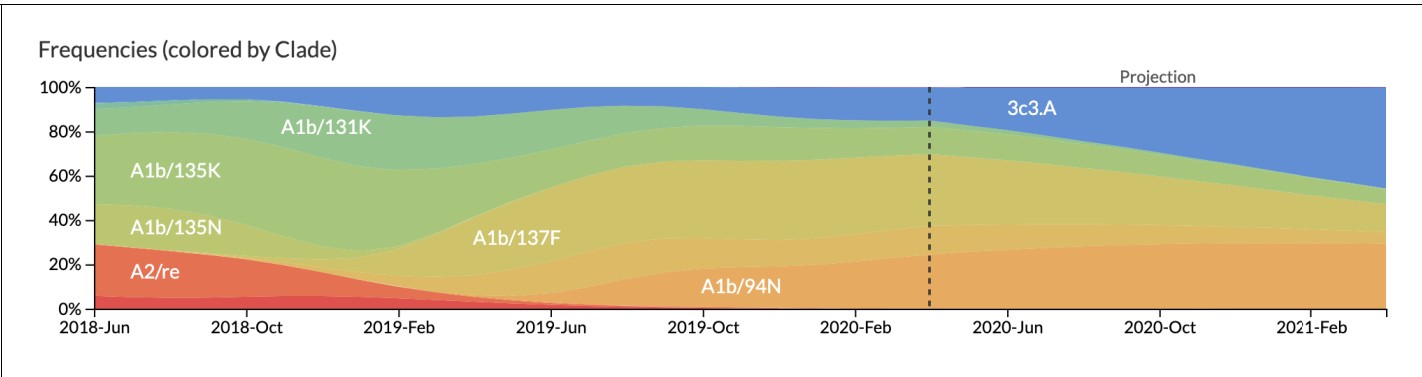

**Figure 12.** Snapshot of live forecasts on nextstrain.org from our best model (HI antigenic novelty and mutational load) for April 1, 2021. The observed frequency trajectories for currently circulating clades are shown up to April 1, 2020. Our model forecasts growth of the clades 3c3.A and A1b/94N and decline of all other major clades.

stochastic clade definitions from phylogenies. The best models from this framework still effectively capture clade dynamics and accurately identify optimal vaccine candidates from simulated and natural H3N2 populations without relying on clades as model targets. We have further introduced novel fitness metrics based on experimental measurements of antigenic drift and functional constraint. We demonstrated that the integration of these phenotypic metrics with previously published sequence-only metrics produces more accurate forecasts than sequence-only models. Interestingly, we found that a naive model that predicts no change over the course of one year can often identify a single representative strain of the future despite its inability to accurately forecast clade frequencies. We have added this framework as a component of seasonal influenza analyses on nextstrain.org where it provides real-time forecasts for influenza researchers, decision makers, and the public.

## Integration of genotypic and phenotypic metrics minimizes overfitting

Our evaluation of models by time-series cross-validation and true out-of-sample forecasts revealed substantial potential for model overfitting. We observed overfitting to both specific genetic backgrounds and general historical contexts. A clear example of the former was the poor performance of our DMS-based fitness metric compared to a simpler mutational load metric. Although the DMS experiments provided detailed estimates of which amino acids were preferred at which positions in HA, these measurements were specific to a single strain, A/Perth/16/2009 (*Lee et al., 2018*). When we applied these measurements to predict the success of global populations, they were less informative on average than the naive model. To benefit from the more comprehensive fitness costs measured by DMS data, future models will need to synthesize DMS measurements across multiple H3N2 strains from distinct genetic contexts. We anticipate that these measurements could be used to define and continually update a modern set of sites contributing to mutational load in natural populations. This set of sites could replace the statically defined set of 'non-epitope' sites we use to estimate mutational load here.

We observed overfitting to historical context in sequence-based models of antigenic drift. The fitness benefit of mutations that led to antigenic drift in H3N2 in the past is well-documented (*Wiley et al., 1981*; *Smith et al., 2004*; *Wolf et al., 2006*; *Koel et al., 2013*). Although the antigenic importance of seven specific sites in HA were experimentally validated by *Koel et al., 2013*, these sites do not explain all antigenic drift observed in natural populations (*Neher et al., 2016*). Other attempts to define these so-called 'epitope sites' have relied on either aggregation of results from antigenic escape assays (*Wolf et al., 2006*) or retrospective computational analyses of sites with beneficial mutations (*Shih et al., 2007*; *Luksza and Lässig, 2014*). We found that models based on all of these definitions except for the seven Koel epitope sites overfit to the historical context from which they were identified (*Table 6*). These results suggest that the set of sites that contribute to antigenic drift at any given time may depend on both the fitness landscape of currently circulating strains and the immune landscape of the hosts these strains need to infect. Recent experimental mapping of antigenic escape mutations in H3N2 HA with human sera show that the specific sites

that confer antigenic escape can vary dramatically between individuals based on their exposure history (*Lee et al., 2019*). In contrast to models based on predefined 'epitope sites', our model based on experimental measurements of antigenic drift did not suffer from overfitting in the validation or test periods. We suspect that this model was able to minimize overfitting by continuously updating its antigenic model with recent experimental data and assigning antigenic weight to branches of a phylogeny rather than specific positions in HA.

Even the most accurate models with few parameters will sometimes fail due to the probabilistic nature of evolution. For example, the model with the best performance across our validation data – mutational load and LBI – was also one of the worst models across our test data. Although we cannot rule out the role of overfitting, this model's poor performance coincided with unusual evolutionary circumstances. The diversity of H3N2 lineages during our test period was higher than the historical average (*Koelle et al., 2006*), with the most recent common ancestor of all circulating strains dating eight years back. This persistence of diversity may have reduced the effectiveness of the LBI metric that assumes relatively rapid population turnover. Additionally, this model's poorest performance occurred in 2019 when it failed to predict the sudden decline of a dominant reassortant clade, A2/re. Only our models based on HI antigenic novelty and mutational load continued to perform as well or better than the naive model during the same time period. These results highlight the challenge of identifying models that remain robust to stochastic evolutionary events by avoiding overfitting to the past.

Correspondingly, we observed that composite models of multiple orthogonal fitness metrics often outperformed models based on their individual components. These results are consistent with previous work that found improved performance by integrating components of antigenic drift, functional constraint, and clade growth (*Luksza and Lässig, 2014*). However, the effective elimination of LBI from our three-metric model during the validation period (*Figure 9*) reveals the limitations of our current additive approach to composite models. The recent success of weighted ensembles for

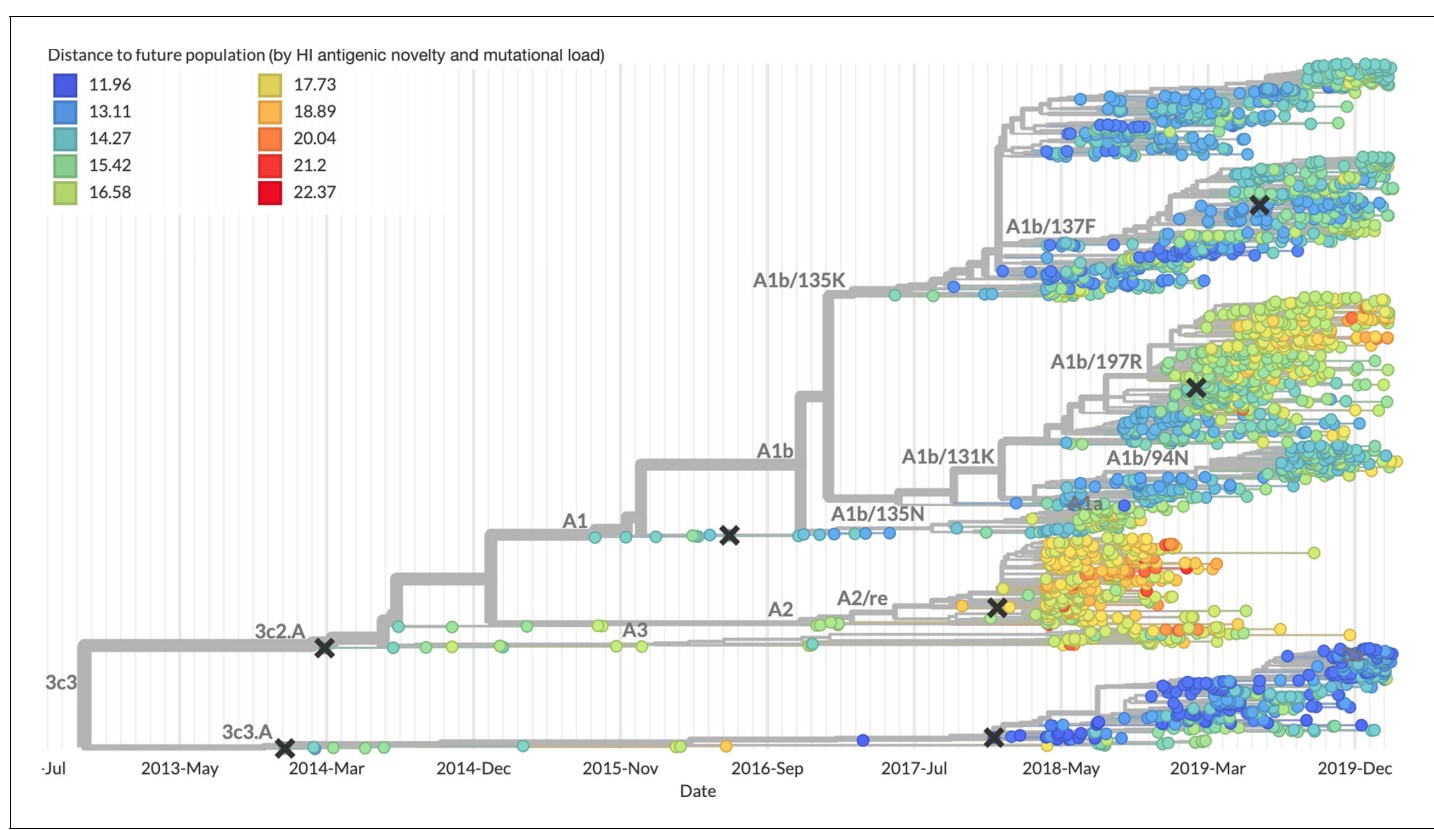

**Figure 13.** Snapshot of the last two years of seasonal influenza H3N2 evolution on nextstrain.org showing the estimated distance per strain to the future population. Distance to the future is calculated for each strain as the Hamming distance of HA amino acid sequences to all other circulating strains weighted by the other strain's projected frequencies under the best fitness model (HI antigenic novelty and mutational load).

short-term influenza forecasting through the CDC's FluSight network (*Reich et al., 2019*) suggests that long-term forecasting may benefit from a similar approach.

## Forecasting framework aids practical forecasts

By forecasting the composition of future H3N2 populations with biologically-informed fitness metrics, our best models consistently outperformed a naive model (*Table 6*). While this performance confirms previously demonstrated potential for long-term influenza forecasting (*Łuksza and Lässig, 2014*), the average gain from these models over the naive model appears low at 0.96 AAs per year for validation data and 0.85 AAs per year for test data. However, these results are consistent with the observed dynamics of H3N2. First, the one-year forecast horizon is a fraction of the average coalescence time for H3N2 populations of about 3–8 years (*Rambaut et al., 2008*). Hence, we expect the diversity of circulating strains to persist between seasons. Second, H3N2 hemagglutinin accumulates 3.6 amino acid changes per year (*Smith et al., 2004*). This accumulation of amino acid substitutions contributes to the distance between annual populations observed by the naive model. In this context, our model gains of 0.96 and 0.85 AAs per year correspond to an explanation of 27% and 24% of the expected additional distance between annual populations, respectively.

Several clear opportunities to improve forecasts still remain. Integration of more recent experimental data may improve estimates of antigenic drift. Despite the weak performance of our FRA antigenic novelty model on recent data, continued accumulation of FRA measurements over time should eventually enable models as accurate as the current HI-based models. In addition to these FRA data based on ferret antisera, recent high-throughput antigenic escape assays with human sera promise to improve existing definitions of epitope sites (*Lee et al., 2019*). These assays reveal the specific sites and residues that confer antigenic escape from polyclonal sera obtained from individual humans. A sufficiently broad geographic and temporal sample of human sera with these assays could reveal consistent patterns of the immune landscape H3N2 strains must navigate to be globally successful. Models should also integrate information from multiple segments of the influenza genome and will need to balance the fitness benefits of evolution in genes such as neuraminidase (*Chen et al., 2018*) with the costs of reassortment (*Villa and Lässig, 2017*). Our forecasting framework makes the inclusion of fitness metrics based on additional gene segments technically straightforward. However, the definition of appropriate fitness metrics for neuraminidase and other genes remains an important scientific challenge. An additional challenge to model training is a relative lack of historical strains for which all genes have been sequenced. Of the 34,312 H3N2 strains in GISAID with all eight primary gene segments and collection dates between October 1, 1990 and 2019, the majority (24,466 or 71%) were collected after October 1, 2015. Data availability will therefore inform which gene segments are prioritized for inclusion in future models. Finally, forecasting models need to account for the geographic distribution of viruses and the vastly different sampling intensities across the globe. Most influenza sequence data come from highly developed countries that account for a small fraction of the global population, while globally successful clades of influenza H3N2 often emerge in less well-sampled regions (*Russell et al., 2008*; *Rambaut et al., 2008*; *Bedford et al., 2015*). Explicitly accounting for these sampling biases and the associated migration dynamics would allow models to weight forecasts based on both viral fitness and transmission.

## The nature of the predictive power of individual metrics remains unclear

Prediction of future influenza virus populations is intrinsically limited by the small number of data points available to train and test models. Increasingly more complex models are therefore prone to overfitting. Across the validation and test periods, we found that antigenic drift and mutational load were the most robust predictors of future success for seasonal influenza H3N2 populations.

Several metrics like the rate of frequency change or epitope mutations are naively expected to have predictive power but do not. Others metrics like the mutational load are not expected to measure adaptation but are predictive. These results point to one aspect that often overlooked when comparing the genetic make-up of an asexual population at two time points: the future population is unlikely to descend from any of the sampled tips but ancestral lineages of the future population merge with those of the present population in the past. Optimal representatives of the future therefore tend to be tips in the present that tend to be basal and less evolved. The LBI and the mutational

load metric have the tendency to assign low fitness to evolved tips. The LBI in particular assigns high fitness to the base of large clades. Much of the predictive power, in the sense of a reduced distance between the predicted and observed populations, might be due to putting more weight on less evolved strains rather than *bona fide* prediction of fitness. In a companion manuscript, *Barrat-Charlaix et al., 2020* show that LBI has little predictive power for fixation probabilities of mutations in H3N2.

Our framework enables real-time practical forecasts of these populations by leveraging historical and modern experimental assays and gene sequences. By releasing our framework as an open source tool based on modern data science standards like tidy data frames, we hope to encourage continued development of this tool by the influenza research community. We additionally anticipate that the ability to forecast the sequence composition of populations with earth mover's distance will enable future forecasting research with pathogens whose genomes cannot be analyzed by traditional phylogenetic methods including recombinant viruses, bacteria, and fungi.

### Model sharing and extensions

The entire workflow for our analyses was implemented with Snakemake (*Köster and Rahmann, 2012*). We have provided all source code, configuration files, and datasets at https://github.com/blab/flu-forecasting (*Huddleston, 2020*; copy archived at https://github.com/elifesciences-publications/flu-forecasting).

## Materials and methods

### Simulation of influenza H3N2-like populations

We simulated the long-term evolution of H3N2-like viruses with SANTA-SIM (*Jariani et al., 2019*) for 10,000 generations or 50 years where 200 generations was equivalent to 1 year. We discarded the first 10 years as a burn-in period, selected the next 30 years for model fitting and validation, and held out the last 9 years as out-of-sample data for model testing (*Figure 2*). Each simulated population was seeded with the full length HA from A/Beijing/32/1992 (NCBI accession: U26830.1) such that all simulated sequences contained signal peptide, HA1, and HA2 domains. We defined purifying selection across all three domains, allowing the preferred amino acid at each site to change at a fixed rate over time. We additionally defined exposure-dependent selection for 49 putative epitope sites in HA1 (*Luksza and Lässig, 2014*) to impose an effect of antigenic novelty that would allow mutations at those sites to increase viral fitness despite underlying purifying selection. We modified the SANTA-SIM source code to enable the inclusion of true fitness values for each strain in the FASTA header of the sampled sequences from each generation. This modified implementation has been integrated into the official SANTA-SIM code repository at https://github.com/santa-dev/santa-sim as of commit e2b3ea3. For our full analysis of model performance, we sampled 90 viruses per month to match the sampling density of natural populations. For tuning of hyperparameters, we sampled 10 viruses per month to enable rapid exploration of hyperparameter space.

### Hyperparameter tuning with simulated populations

To avoid overfitting our models to the relatively limited data from natural populations, we used simulated H3N2-like populations to tune hyperparameters including the KDE bandwidth for frequency estimates and the L1 penalty for model coefficients. We simulated populations, as described above, and fit models for each parameter value using the true fitness of strains from the simulator.

We identified the optimal KDE bandwidth for frequencies as the value that minimized the difference between the mean distances to the future from the true fitness model and the naive model. We set the L1 lambda penalty to zero, to reduce variables in the analysis and avoid interactions between the coefficients and the KDE bandwidths. Higher bandwidths completely wash out dynamics of populations by making all strains appear to exist for long time periods. This flattening of frequency trajectories means that as bandwidths increase, the naive model gets more accurate and less informative. Given this behavior, we found the bandwidth that produced the minimum difference between distances to the future for the true fitness and naive models instead of the bandwidth that produced the minimum mean model distance. Based on this analysis, we identified an optimal bandwidth of $\frac{2}{12}$ or the equivalent of 2 months for floating point dates. Next, we identified an L1 penalty

of 0.1 for model coefficients that minimized the mean distance to the future for the true fitness model.

## Antigenic data

Hemagglutination inhibition (HI) and focus reduction assay (FRA) measurements were provided by WHO Global Influenza Surveillance and Response System (GISRS) Collaborating Centers in London, Melbourne, Atlanta and Tokyo. We converted these raw two-fold dilution measurements to $log_2$ titer drops normalized by the corresponding $log_2$ autologous measurements as previously described (*Neher et al., 2016*).

## Strain selection for natural populations

Prior to our analyses, we downloaded all HA sequences and metadata from GISAID (*Shu and McCauley, 2017*). For model training and validation, we selected 15,583 HA sequences $\geq$ 900 nucleotides that were sampled between October 1, 1990 and October 1, 2015. To account for known variation in sequence availability by region, we subsampled the selected sequences to a representative set of 90 viruses per month with even sampling across 10 global regions including Africa, Europe, North America, China, South Asia, Japan and Korea, Oceania, South America, Southeast Asia, and West Asia. We excluded all egg-passaged strains and all strains with ambiguous year, month, and day annotations. We prioritized strains with more available HI titer measurements provided by the WHO GISRS Collaborating Centers. For model testing, we selected an additional 7,171 HA sequences corresponding to 90 viruses per month sampled between October 1, 2015 and October 1, 2019. We used these test sequences to evaluate the out-of-sample error of fixed model parameters learned during training and validation. *Supplementary file 1* describes contributing laboratories for all 22,754 validation and test strains.

## Phylogenetic inference

For each timepoint in model training, validation, and testing, we selected the subsampled HA sequences with collection dates up to that timepoint. We aligned sequences with the augur align command (*Hadfield et al., 2018*) and MAFFT v7.407 (*Katoh et al., 2002*). We inferred initial phylogenies for HA sequences at each timepoint with IQ-TREE v1.6.10 (*Nguyen et al., 2015*). To reconstruct time-resolved phylogenies, we applied TreeTime v0.5.6 (*Sagulenko et al., 2018*) with the augur refine command.

## Frequency estimation

To account for uncertainty in collection date and sampling error, we applied a kernel density estimation (KDE) approach to calculate global strain frequencies. Specifically, we constructed a Gaussian kernel for each strain with the mean at the reported collection date and a variance (or KDE bandwidth) of two months. The bandwidth was identified by cross-validation, as described above. This bandwidth also roughly corresponds to the median lag time between strain collection and submission to the GISAID database. We estimated the frequency of each strain at each timepoint by calculating the probability density function of each KDE at that timepoint and normalizing the resulting values to sum to one. We implemented this frequency estimation logic in the augur frequencies command.

## Model fitting and evaluation

### Fitness model

We assumed that the evolution seasonal influenza H3N2 populations can be represented by a Malthusian growth fitness model, as previously described (*Luksza and Lässig, 2014*). Under this model, we estimated the future frequency, $\hat{x}_i(t + \Delta t)$, of each strain $i$ from the strain's current frequency, $x_i(t)$, and fitness, $f_i(t)$, as follows where the resulting future frequencies were normalized to one by $\frac{1}{Z(t)}$.

$$\hat{x}_i(t + \Delta t) = \frac{1}{Z(t)} x_i(t) \exp(f_i(t)\Delta t) \tag{1}$$

We defined the fitness of each strain at time *t* as the additive combination of one or more fitness

metrics, $f_{i,m}$, scaled by fitness coefficients, $\beta_m$. For example, *Equation 2* estimates fitness per strain by mutational load (*ml*) and local branching index (*lbi*).

$$f_i(t) = \beta_{\text{ne}}f_{i,\text{ml}}(t) + \beta_{\text{lbi}}f_{i,\text{lbi}}(t) \tag{2}$$

## Model target

For a model based on any given combination of fitness metrics, we found the fitness coefficients that minimized the earth mover's distance (EMD) (*Rubner et al., 1998*; *Kusner et al., 2015*) between amino acid sequences from the observed future population at time $u = t + \Delta t$ and the estimated future population created by projecting frequencies of strains at time $t$ by their estimated fitnesses. Solving for EMD identifies the minimum about of 'earth' that must be moved from a source population to a sink population to make those populations as similar as possible. This solution requires both a 'ground distance' between pairs of strains from both populations and weights assigned to each strain that determine how much that strain contributes to the overall distance.

For each timepoint $t$ and corresponding timepoint $u = t + 1$, we defined the ground distance as the Hamming distance between HA amino acid sequences for all pairs of strains between timepoints. For strains with less than full length nucleotide sequences, we inferred missing nucleotides through TreeTime's ancestral sequence reconstruction analysis. We defined weights for strains at timepoint $t$ based on their projected future frequencies. We defined weights for strains at timepoint $u$ based on their observed frequencies. We then identified the fitness coefficients that provided projected future frequencies that minimized the EMD between the estimated and observed future populations. With this metric, a perfect estimate of the future's strain sequence composition and frequencies would produce a distance of zero. However, the inevitable accumulation of substitutions between the two populations prevents this outcome. We calculated EMD with the Python bindings for the OpenCV 3.4.1 implementation (*Bradski, 2000*). We applied the Nelder-Mead minimization algorithm as implemented in SciPy (*Virtanen et al., 2020*) to learn fitness coefficients that minimize the average of this distance metric over all timepoints in a given training window.

## Lower bound on earth mover's distance

The minimum distance to the future between any two timepoints cannot be zero due to the accumulation of mutations between populations. We estimated the lower bound on earth mover's distance between timepoints using the following greedy solution to the optimal transport problem. For each timepoint $t$, we initialized the optimal frequency of each current strain to zero. For each strain in the future timepoint $u$, we identified the closest strain in the current timepoint by Hamming distance and added the frequency of the future strain to the optimal frequency of the corresponding current strain. This approach allows each strain from timepoint $t$ to accumulate frequencies from multiple strains at timepoint $u$. We calculated the minimum distance between populations as the earth mover's distance between the resulting optimal frequencies for current strains, the observed frequencies of future strains, and the original distance matrix between those two populations.

## Strain-specific distance to the future

We calculated the weighted Hamming distance to the future of each strain from the strain's HA amino acid sequence and the frequencies and sequences of the corresponding population one year in the future. Specifically, the distance between any strain $i$ from timepoint $t$ to the future timepoint $u$ was the Hamming distance, $h$, between strain $i$'s amino acid sequence, $s_i$, each future strain $j$'s amino acid sequence, $s_j$, and the frequency of strain $j$ in the future timepoint, $x_j(u)$.

$$d_i(u) = \sum_{j \in s(u)} x_j(u)h(s_i, s_j) \tag{3}$$

We calculated the estimated distance to the future for live forecasts with the same approach, replacing the observed future population frequencies and sequences with the estimated population based on our models.

$$d_i(\hat{u}) = \sum_{j \in s(\hat{u})} x_j(\hat{u}) h(s_i, s_j) \tag{4}$$

## Time-series cross-validation

To obtain unbiased estimates for the out-of-sample errors of our models, we adopted the standard cross-validation strategy of training, validation, and testing. We divided our available data into an initial training and validation set spanning October 1990 to October 2015 and an additional testing set spanning October 2015 to October 2019 (*Figure 7*). We partitioned our training and validation data into six month seasons corresponding to winter in the Northern Hemisphere (October–April) and the Southern Hemisphere (April–October) and trained models to estimate frequencies of populations one year into the future from each season in six-year sliding windows. To calculate validation error for each training window, we applied the resulting model coefficients to estimate the future frequencies for the year after the last timepoint in the training window. These validation errors informed our tuning of hyperparameters. Finally, we fixed the coefficients for each model at the mean values across all training windows and applied these fixed models to the test data to estimate the true forecasting accuracy of each model on previously unobserved data.

## Model comparison by bootstrap tests

We compared the performance of different pairs of models using bootstrap tests. For each timepoint, we calculated the difference between one model's earth mover's distance to the future and the other model's distance. Values less than zero in the resulting empirical distribution represent when the first model outperformed the second model. To determine whether the first model generally outperformed the second model, we bootstrapped the empirical difference distributions for n = 10,000 samples and calculated the mean difference of each bootstrap sample. We calculated an empirical p value for the first model as the proportion of bootstrap samples with mean values greater than or equal to zero. This p value represents how likely the mean difference between the models' distances to the future is to be zero or greater. We measured the effect size of each comparison as the mean ±the standard deviation of the bootstrap distributions. We performed pairwise model comparisons for all biologically-informed models against the naive model (*Figures 14* and *15*). We also compared a subset of composite models to their respective individual models (*Table 5*).

## Fitness metrics

We defined the following fitness metrics per strain and timepoint.

## Antigenic drift

We estimated antigenic drift for each strain using either genetic or HI data. To estimate antigenic drift with genetic data, we implemented an antigenic novelty metric based on the 'cross-immunity' metric originally defined by *Luksza and Lässig, 2014*. Briefly, for each pair of strains in adjacent seasons, we counted the number of amino acid differences between the strains' HA sequences at 49 epitope sites. The one-based coordinates of these sites relative to the start of the HA1 segment were 50, 53, 54, 121, 122, 124, 126, 131, 133, 135, 137, 142, 143, 144, 145, 146, 155, 156, 157, 158, 159, 160, 163, 164, 172, 173, 174, 186, 188, 189, 190, 192, 193, 196, 197, 201, 207, 213, 217, 226, 227, 242, 244, 248, 275, 276, 278, 299, and 307. We limited pairwise comparisons to all strains sampled within the last five years from each timepoint. For each individual strain *i* at each timepoint *t*, we estimated that strain's ability to escape cross-immunity by summing the exponentially-scaled epitope distances between previously circulating strains and the given strain as in *Equation 5*. We defined the constant $D_0 = 14$, as in the original definition of cross-immunity (*Luksza and Lässig, 2014*). To compare these epitope sites with other previously published sites, we fit epitope antigenic novelty models based on sites defined by *Wolf et al., 2006* and *Koel et al., 2013*.

$$f_{i,\text{ep}}(t) = \sum_{j:t_j < t_i} -\max(x_j) \exp\left(-D_{\text{ep}}(a_i, a_j)/D_0\right) \tag{5}$$

To test the historical contingency of the epitope sites defined above, we additionally identified a

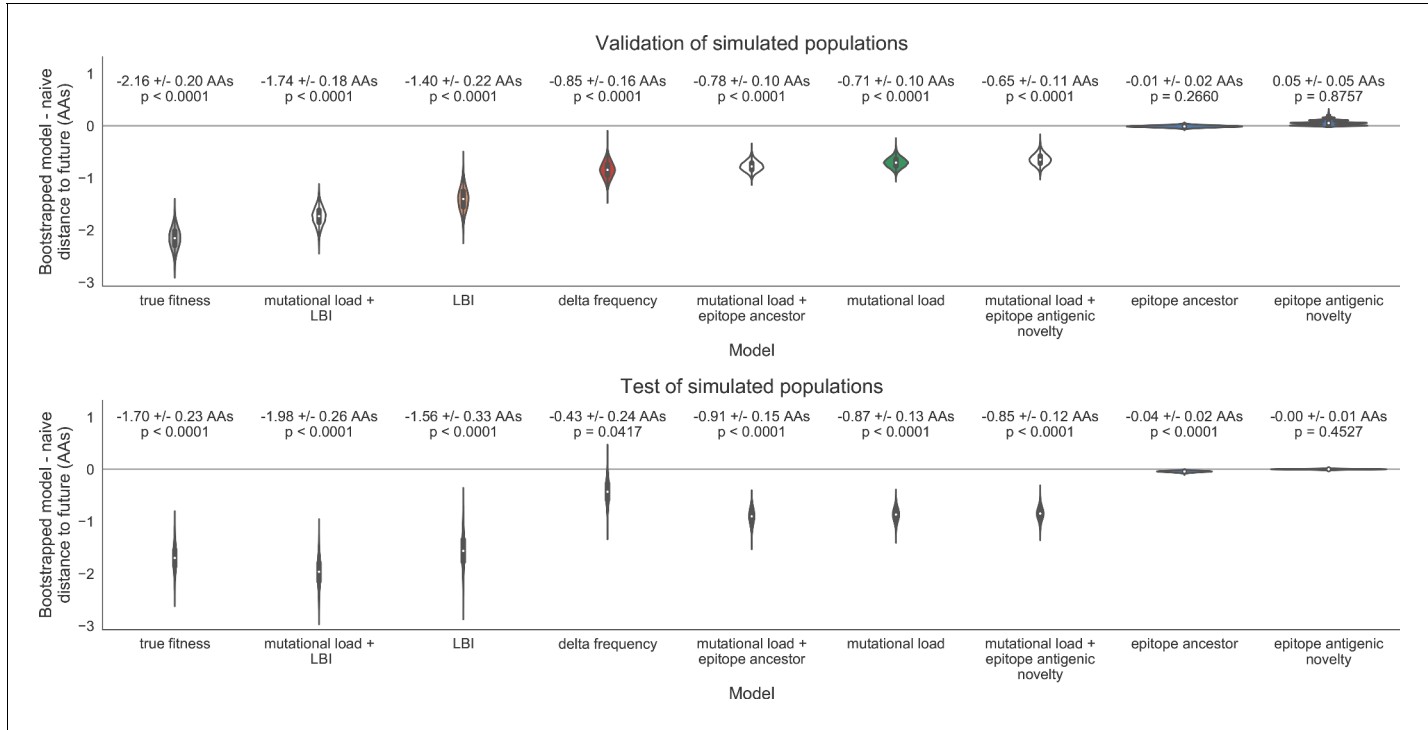

**Figure 14.** Bootstrap distributions of the mean difference of distances to the future between biologically-informed and naive models for simulated populations. Empirical differences in distances to the future were sampled with replacement and mean values for each bootstrap sample were calculated across n = 10,000 bootstrap iterations. The horizontal gray line indicates a difference of zero between a given model and its corresponding naive model. Each model is annotated by the mean ± the standard deviation of the bootstrap distribution. Models are also annotated by the p-value representing the proportion of bootstrap samples with values less than zero (see Methods).

new set of sites with beneficial mutations across the training/validation period of October 1990 through October 2015. Following the general approach of *Shih et al., 2007*, we manually identified 25 sites in HA1 where mutations rapidly swept through the global population. We required mutations to emerge from below 5% global frequency and reach > 90% frequency. Although we did not require sweeps to complete within a fixed amount of time, we observed that they required no longer than one to three years to complete. To minimize false positives, we eliminated any sites where one or more mutations rose above 20% frequency and subsequently died out. If two or more sites had redundant sweep dynamics (mutations emerging and fixing at the same times), we retained the site with the most mutational sweeps. Based on this requirements, we defined our final collection of 'oracle' sites in HA1 coordinates as 3, 45, 48, 50, 75, 140, 145, 156, 158, 159, 173, 186, 189, 193, 198, 202, 212, 222, 223, 225, 226, 227, 278, 311, and 312.

To estimate antigenic drift with HI data, we first applied the titer tree model to the phylogeny at a given timepoint and the corresponding HI data for its strains, as previously described by *Neher et al., 2016*. This method effectively estimates the antigenic drift per branch in units of $log_2$ titer change. We selected all strains with nonzero frequencies in the last six months as 'current strains' and all strains sampled five years prior to that threshold as 'past strains'. Next, we calculated the pairwise antigenic distance between all current and past strains as the sum of antigenic drift weights per branch on the phylogenetic path between each pair of strains. Finally, we calculated each strain's ability to escape cross-immunity using *Equation 5* with the pairwise distances between epitope sequences replaced with pairwise antigenic distance from HI data. As with the original epitope antigenic novelty described above, this HI antigenic novelty metric produces higher values for strains that are more antigenically distinct from previously circulating strains.

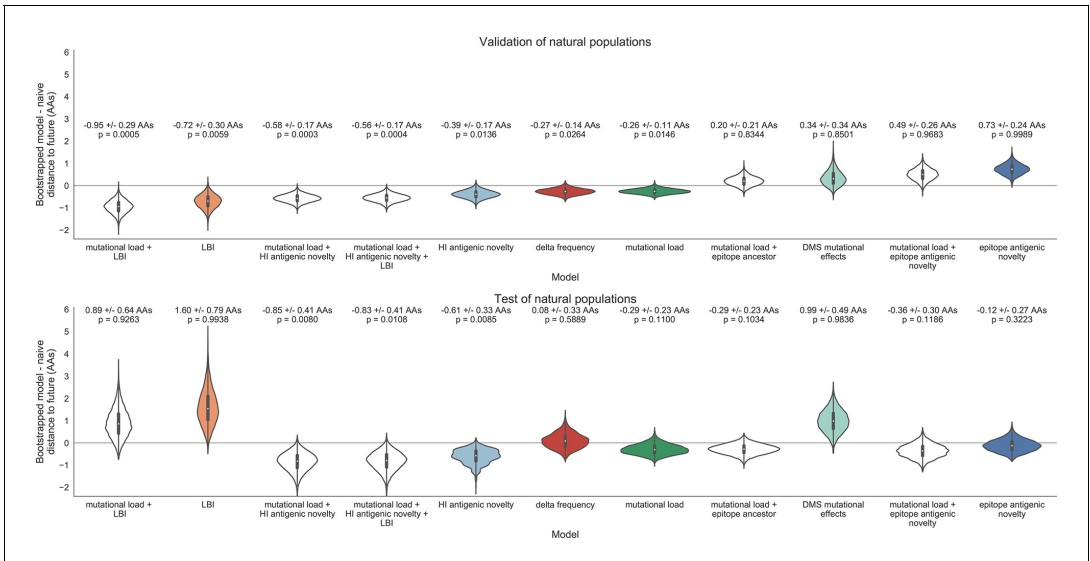

**Figure 15.** Bootstrap distributions of the mean difference of distances to the future between biologically-informed and naive models for natural populations. Empirical differences in distances to the future were sampled with replacement and mean values for each bootstrap sample were calculated across n = 10,000 bootstrap iterations. The horizontal gray line indicates a difference of zero between a given model and its corresponding naive model. Each model is annotated by the mean ± the standard deviation of the bootstrap distribution. Models are also annotated by the p-value representing the proportion of bootstrap samples with values less than zero (see Materials and methods).

## Functional constraint

We estimated functional constraint for each strain using either genetic or deep mutational scanning (DMS) data. To estimate functional constraint with genetic data, we implemented the non-epitope mutation metric originally defined by *Luksza and Lässig, 2014*. This metric counts the number of amino acid differences at 517 non-epitope sites in HA sequences between each strain $i$ at timepoint $t$ and that strain's most recent inferred ancestral sequence in the previous season ($t - 1$).

We estimated functional constraint using mutational preferences from DMS data as previously defined (*Lee et al., 2018*). Briefly, mutational effects were defined as the log ratio of DMS preferences, $\pi$, at site $r$ for the derived amino acid, $a_i$, and the ancestral amino acid, $a_j$. As with the non-epitope mutation metric above, we considered only substitutions in HA between each strain $i$ and that strain's most recent inferred ancestral sequence in the previous season. We calculated the total effect of these substitutions as the sum of the mutational preferences for each substitution, as in *Equation 6*.

$$f_{i,\mathrm{DMS}}(t) = \sum_{r \in r, a_i != r, a_j} \log_2 \frac{\pi_{r,a_i}}{\pi_{r,a_j}} \tag{6}$$

To determine whether DMS preferences could be used to define fitness metrics that were less dependent on the historical context of the background strain, we implemented two additional DMS-based metrics: 'DMS entropy' and 'DMS mutational load'. For both metrics, we calculated the distance between HA amino acid sequences of each strain and its ancestral sequence in the previous season, to enable comparison of these metrics with the DMS mutational effects and mutational load metrics. For the 'DMS entropy' metric, we calculated the distance between sequences such that each mismatch was weighted by the inverse entropy of DMS preferences at the site of the mismatch. We expected this metric to produce a negative coefficient similar to the mutational load metric, as higher values will result from mutations at sites with lower entropy and, thus, lower tolerance for mutations. For the 'DMS mutational load' metric, we defined a novel set of non-epitope sites corresponding to each position in HA with a standardized entropy less than zero. With this metric, we sought to identify more highly conserved sites without weighting any one site differently from others. We anticipated that this lack of site-specific weighting would make the DMS mutational load metric even less background-dependent than the DMS entropy and DMS mutational effect metrics.

## Clade growth

We estimated clade growth for each strain using local branching index (LBI) and the change in frequency over time (delta frequency). To calculate LBI for each strain at each timepoint, we applied the LBI heuristic algorithm as originally described (*Neher et al., 2014*) to the phylogenetic tree constructed at each timepoint. We set the neighborhood parameter, $\tau$, to 0.3 and only considered viruses sampled in the last 6 months of each phylogeny as contributing to recent clade growth.

We estimated the change in frequency over time by calculating clade frequencies under a Brownian motion diffusion process as previously described (*Lee et al., 2018*). These frequency calculations allowed us to assign a partial clade frequency to each strain within nested clades. We calculated the delta frequency as the change in frequency for each strain between the most recent timepoint in a given phylogeny and six months prior to that timepoint divided by 0.5 years.

For the purpose of visualizing related amino acid sequences in *Figure 1*, we applied dimensionality reduction to pairwise amino acid distances followed by hierarchical clustering. Specifically, we selected a representative tree from our simulated population of viruses at month 10 of year 30. From this tree, we selected all strains with a collection date in the previous two years. We calculated the pairwise Hamming distance between the full-length HA amino acid sequences for all selected strains and applied t-SNE dimensionality reduction (*van der Maaten and Hinton, 2008*) to the resulting distance matrix (n = 2 components, perplexity = 30.0, and learning rate = 400). We assigned each strain to a cluster based on its two-dimensional t-SNE embedding using DBSCAN (*Ester et al., 1996*) with a maximum neighborhood distance of 10 AAs and a minimum of 20 strains per cluster. Despite known limitations of applying hierarchical clustering to manifold projections that do not preserve sample density, this approach allowed us to effectively assign strains to qualitative genetic clusters for the purposes of visualization.

## Data and software availability

Sequence data are available from GISAID using accession ids provided in *Supplementary file 1*. Source code, derived data from serological measurements, fitness metric annotations, and resulting fitness model performance data are available in the project's GitHub repository (*Huddleston, 2020*; copy archived at https://github.com/elifesciences-publications/flu-forecasting). Raw serological measurements are restricted from public distribution by previous data sharing agreements.

## Acknowledgements

We thank the Influenza Division at the US Centers for Disease Control and Prevention, the Victorian Infectious Diseases Reference Laboratory at the Australian Peter Doherty Institute for Infection and Immunity, the Influenza Virus Research Center at the Japan National Institute of Infectious Diseases, the Crick Worldwide Influenza Centre at the UK Francis Crick Institute for sharing HI and FRA data.

We gratefully acknowledge the authors, originating and submitting laboratories of the sequences from the GISAID EpiFlu Database (*Shu and McCauley, 2017*) on which this research is based. The list is detailed in the Supplemental Material.

We thank Jesse Bloom, Erick Matsen, Bing Brunton, Harmit Malik, Sidney Bell, Allison Black, Lola Arakaki, Duncan Ralph, and members of the Bedford lab for useful advice and discussions. JH is a Graduate Research Fellow and is supported by the NIH grant NIAID F31AI140714. The work done at the Crick Worldwide Influenza Centre was supported by the Francis Crick Institute receiving core funding from Cancer Research UK (FC001030), the Medical Research Council (FC001030) and the Wellcome Trust (FC001030). SF, KN, NK, SW and HH were supported by the Ministry of Health, Labour and Welfare, Japan (10110400). SW was supported by the Japan Agency for Medical Research and Development (JPfk0108118). The Melbourne WHO Collaborating Centre for Reference and Research on Influenza is supported by the Australian Government Department of Health. PB and RAN are supported by NIAID R01 AI127893-01 and institutional core funding. TB is a Pew Biomedical Scholar and is supported by NIH grants NIGMS R35 GM119774-01, NIAID U19 AI117891-01 and NIAID R01 AI127893-01.

The findings and conclusions in this report are those of the author(s) and do not necessarily represent the official position of the Centers for Disease Control and Prevention.

# Additional information

## Funding

| Funder | Grant reference number | Author |
| --- | --- | --- |
| Cancer Research UK | FC001030 | Lynne Whittaker<br>Burcu Ermetal<br>Rodney Stuart Daniels<br>John W McCauley |
| Medical Research Council | FC001030 | Lynne Whittaker<br>Burcu Ermetal<br>Rodney Stuart Daniels<br>John W McCauley |
| Wellcome | FC001030 | Lynne Whittaker<br>Burcu Ermetal<br>Rodney Stuart Daniels<br>John W McCauley |
| Ministry of Health, Labour and Welfare | 10110400 | Seiichiro Fujisaki<br>Kazuya Nakamura<br>Noriko Kishida<br>Shinji Watanabe<br>Hideki Hasegawa |
| Japan Agency for Medical Research and Development | JPfk0108118 | Shinji Watanabe |
| Australian Government Department of Health | | Ian Barr<br>Kanta Subbarao |
| National Institute of Allergy and Infectious Diseases | F31AI140714 | John Huddleston |
| National Institute of General Medical Sciences | R35GM119774-01 | Trevor Bedford |
| Pew Charitable Trusts | | Trevor Bedford |
| National Institute of Allergy and Infectious Diseases | U19AI117891-01 | Trevor Bedford |
| National Institute of Allergy and Infectious Diseases | R01AI127893-01 | Pierre Barrat-Charlaix<br>Richard A Neher<br>Trevor Bedford |

The funders had no role in study design, data collection and interpretation, or the decision to submit the work for publication.

## Author contributions

John Huddleston, Conceptualization, Data curation, Software, Formal analysis, Funding acquisition, Validation, Investigation, Visualization, Methodology, Writing - original draft, Project administration, Writing - review and editing; John R Barnes, Thomas Rowe, Xiyan Xu, Rebecca Kondor, David E Wentworth, Lynne Whittaker, Burcu Ermetal, Rodney Stuart Daniels, John W McCauley, Seiichiro Fujisaki, Kazuya Nakamura, Noriko Kishida, Shinji Watanabe, Hideki Hasegawa, Ian Barr, Resources, Investigation; Kanta Subbarao, Conceptualization, Resources, Supervision, Investigation, Methodology, Writing - review and editing; Pierre Barrat-Charlaix, Richard A Neher, Conceptualization, Software, Supervision, Funding acquisition, Methodology, Project administration, Writing - review and editing; Trevor Bedford, Conceptualization, Resources, Software, Supervision, Funding acquisition, Investigation, Methodology, Project administration, Writing - review and editing

## Author ORCIDs

John Huddleston https://orcid.org/0000-0002-4250-2063
Rebecca Kondor http://orcid.org/0000-0002-2596-4282
David E Wentworth http://orcid.org/0000-0002-5190-980X
John W McCauley http://orcid.org/0000-0002-4744-6347

Kanta Subbarao http://orcid.org/0000-0003-1713-3056
Pierre Barrat-Charlaix http://orcid.org/0000-0002-3816-3724
Richard A Neher http://orcid.org/0000-0003-2525-1407
Trevor Bedford https://orcid.org/0000-0002-4039-5794

### Decision letter and Author response
Decision letter https://doi.org/10.7554/eLife.60067.sa1
Author response https://doi.org/10.7554/eLife.60067.sa2

## Additional files

### Supplementary files
• Supplementary file 1. GISAID accessions and metadata including originating and submitting labs for natural strains used across all timepoints.

• Transparent reporting form

### Data availability
Sequence data are available from GISAID using accession ids provided in Supplementary file 1. Source code, derived data from serological measurements, fitness metric annotations, and resulting fitness model performance data are available in the project's GitHub repository (https://github.com/blab/flu-forecasting; copy archived at https://github.com/elifesciences-publications/flu-forecasting). Raw serological measurements are restricted from public distribution by previous data sharing agreements.

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

## Appendix 1

## GISAID Acknowledgments

WHO Collaborating Centre for Reference and Research on Influenza, Victorian Infectious Diseases Reference Laboratory, Australia; WHO Collaborating Centre for Reference and Research on Influenza, Chinese National Influenza Center, China; WHO Collaborating Centre for Reference and Research on Influenza, National Institute of Infectious Diseases, Japan; The Crick Worldwide Influenza Centre, The Francis Crick Institute, United Kingdom; WHO Collaborating Centre for the Surveillance, Epidemiology and Control of Influenza, Centers for Disease Control and Prevention, United States; ADImmune Corporation, Taiwan; ADPH Bureau of Clinical Laboratories, United States; Aichi Prefectural Institute of Public Health, Japan; Akershus University Hospital, Norway; Akita Research Center for Public Health and Environment, Japan; Alabama State Laboratory, United States; Alaska State Public Health Laboratory, United States; Alaska State Virology Lab, United States; Aomori Prefectural Institute of Public Health and Environment, Japan; Aristotelian University of Thessaloniki, Greece; Arizona Department of Health Services, United States; Arkansas Children's Hospital, United States; Arkansas Department of Health, United States; Auckland Healthcare, New Zealand; Auckland Hospital, New Zealand; Austin Health, Australia; Baylor College of Medicine, United States; California Department of Health Services, United States; Canberra Hospital, Australia; Cantacuzino Institute, Romania; Canterbury Health Services, New Zealand; Caribbean Epidemiology Center, Trinidad and Tobago; CDC GAP Nigeria, Nigeria; CDC-Kenya, Kenya; CEMIC University Hospital, Argentina; CENETROP, Bolivia, Plurinationial State of; Center for Disease Control, Taiwan; Center for Public Health and Environment, Hiroshima Prefectural Technology Research Institute, Japan; Central Health Laboratory, Mauritius; Central Laboratory of Public Health, Paraguay; Central Public Health Laboratory, Ministry of Health, Oman; Central Public Health Laboratory, Palestinian Territory; Central Public Health Laboratory, Papua New Guinea; Central Research Institute for Epidemiology, Russian Federation; Centre for Diseases Control and Prevention, Armenia; Centre for Infections, Health Protection Agency, United Kingdom; Centre Pasteur du Cameroun, Cameroon; Chiba City Institute of Health and Environment, Japan; Chiba Prefectural Institute of Public Health, Japan; Childrens Hospital Westmead, Australia; Chuuk State Hospital, Micronesia, Federated States of; City of El Paso Dept of Public Health, United States; Clinical Virology Unit, CDIM, Australia; Colorado Department of Health Lab, United States; Connecticut Department. of Public Health, United States; Contiguo a Hospital Rosales, El Salvador; Croatian Institute of Public Health , Croatia; CRR virus Influenza region Sud, France; CRR virus Influenza region Sud, Guyana; CSL Ltd, United States; Dallas County Health and Human Services, United States; DC Public Health Lab, United States; Delaware Public Health Lab, United States; Departamento de Laboratorio de Salud Publica, Uruguay; Department of Virology, Medical University Vienna, Austria; Disease Investigation Centre Wates (BBVW), Australia; Drammen Hospital/Vestreviken HF, Norway; Ehime Prefecture Institute of Public Health and Environmental Science, Japan; Erasmus Medical Center, Netherlands; Erasmus University of Rotterdam, Netherlands; Ethiopian Health and Nutrition Research Institute (EHNRI), Ethiopia; Evanston Hospital and North-Shore University, United States; Facultad de Medicina, Spain; Fiji Centre for Communicable Disease Control, Fiji; Florida Department of Health, United States; Fukui Prefectural Institute of Public Health, Japan; Fukuoka City Institute for Hygiene and the Environment, Japan; Fukuoka Institute of Public Health and Environmental Sciences, Japan; Fukushima Prefectural Institute of Public Health, Japan; Gart Naval General Hospital, United Kingdom; Georgia Public Health Laboratory, United States; Gifu Municipal Institute of Public Health, Japan; Gifu Prefectural Institute of Health and Environmental Sciences, Japan; Government Virus Unit, Hong Kong; Gunma Prefectural Institute of Public Health and Environmental Sciences, Japan; Hamamatsu City Health Environment Research Center, Japan; Haukeland University Hospital, Dept. of Microbiology , Norway; Headquarters British Gurkhas Nepal, Nepal; Health Forde, Department of Microbiology, Norway; Health Protection Agency, United Kingdom; Health Protection Inspectorate, Estonia; Hellenic Pasteur Institute, Greece; Hiroshima City Institute of Public Health, Japan; Hokkaido Institute of Public Health, Japan; Hopital Cantonal Universitaire de Geneves, Switzerland; Hopital Charles Nicolle, Tunisia; Hospital Clinic de Barcelona, Spain; Hospital Universitari Vall d'Hebron, Spain; Houston Department of Health and Human Services, United States; Hyogo Prefectural Institute of Public Health and Consumer Sciences, Japan; Ibaraki Prefectural Institute of Public Health, Japan; Illinois Department of Public Health,

United States; Indiana State Department of Health Laboratories, United States; Infectology Center of Latvia, Latvia; Innlandet Hospital Trust, Division Lillehammer, Department for Microbiology, Norway; INSA National Institute of Health Portugal, Portugal; Institut National d'Hygiene, Morocco; Institut Pasteur d'Algerie, Algeria; Institut Pasteur de Dakar, Senegal; Institut Pasteur de Madagascar, Madagascar; Institut Pasteur in Cambodia, Cambodia; Institut Pasteur New Caledonia, New Caledonia; Institut Pasteur, France; Institut Pasteur, Saudi Arabia; Institut Penyelidikan Perubatan, Malaysia; Institute National D'Hygiene, Togo; Institute of Environmental Science and Research, New Zealand; Institute of Environmental Science and Research, Tonga; Institute of Epidemiology and Infectious Diseases, Ukraine; Institute of Epidemiology Disease Control and Research, Bangladesh; Institute of Immunology and Virology Torlak, Serbia; Institute of Medical and Veterinary Science (IMVS), Australia; Institute of Public Health, Serbia; Institute of Public Health, Albania; Institute of Public Health, Montenegro; Institute Pasteur du Cambodia, Cambodia; Instituto Adolfo Lutz, Brazil; Instituto Conmemorativo Gorgas de Estudios de la Salud, Panama; Instituto de Salud Carlos III, Spain; Instituto de Salud Publica de Chile, Chile; Instituto Nacional de Enfermedades Infecciosas, Argentina; Instituto Nacional de Higiene Rafael Rangel, Venezuela, Bolivia; Instituto Nacional de Laboratoriosde Salud (INLASA), Bolivia; Instituto Nacional de Salud de Columbia, Colombia; Instituto Nacional de Saude, Portugal; Iowa State Hygienic Laboratory, United States; IRSS, Burkina Faso; Ishikawa Prefectural Institute of Public Health and Environmental Science, Japan; ISS, Italy; Istanbul University, Turkey; Istituto Superiore di Sanità, Italy; Ivanovsky Research Institute of Virology RAMS, Russian Federation; Jiangsu Provincial Center for Disease Control and Prevention, China; John Hunter Hospital, Australia; Kagawa Prefectural Research Institute for Environmental Sciences and Public Health, Japan; Kagoshima Prefectural Institute for Environmental Research and Public Health, Japan; Kanagawa Prefectural Institute of Public Health, Japan; Kansas Department of Health and Environment, United States; Kawasaki City Institute of Public Health , Japan; Kentucky Division of Laboratory Services, United States; Kitakyusyu City Institute of Enviromental Sciences, Japan; Kobe Institute of Health, Japan; Kochi Public Health and Sanitation Institute, Japan; Kumamoto City Environmental Research Center, Japan; Kumamoto Prefectural Institute of Public Health and Environmental Science, Japan; Kyoto City Institute of Health and Environmental Sciences, Japan; Kyoto Prefectural Institute of Public Health and Environment, Japan; Laboratoire National de Sante Publique, Haiti; Laboratoire National de Sante, Luxembourg; Laboratório Central do Estado do Paraná, Brazil; Laboratorio Central do Estado do Rio de Janeiro, Brazil; Laboratorio de Investigacion/Centro de Educacion Medica y Amistad Dominico Japones (CEMADOJA), Dominican Republic; Laboratorio De Saude Publico, Macao; Laboratorio de Virologia, Direccion de Microbiologia, Nicaragua; Laboratorio de Virus Respiratorio, Mexico; Laboratorio Nacional de Influenza, Costa Rica; Laboratorio Nacional De Salud Guatemala, Guatemala; Laboratorio Nacional de Virologia, Honduras; Laboratory Directorate, Jordan; Laboratory for Virology, National Institute of Public Health, Slovenia; Laboratory of Influenza and ILI, Belarus; LACEN/RS - Laboratório Central de Saúde Pública do Rio Grande do Sul, Brazil; Landspitali - University Hospital, Iceland; Lithuanian AIDS Center Laboratory, Lithuania; Los Angeles Quarantine Station, CDC Quarantine Epidemiology and Surveillance Team, United States; Louisiana Department of Health and Hospitals, United States; Maine Health and Environmental Testing Laboratory, United States; Malbran, Argentina; Marshfield Clinic Research Foundation, United States; Maryland Department of Health and Mental Hygiene, United States; Massachusetts Department of Public Health, United States; Mater Dei Hospital, Malta; Medical Research Institute, Sri Lanka; Medical University Vienna, Austria; Melbourne Pathology, Australia; Michigan Department of Community Health, United States; Mie Prefecture Health and Environment Research Institute, Japan; Mikrobiologisk laboratorium, Sykehuset i Vestfold, Norway; Ministry of Health and Population, Egypt; Ministry of Health of Ukraine, Ukraine; Ministry of Health, Bahrain; Ministry of Health, Kiribati; Ministry of Health, Lao, People's Democratic Republic; Ministry of Health, NIHRD, Indonesia; Ministry of Health, Oman; Minnesota Department of Health, United States; Mississippi Public Health Laboratory, United States; Missouri Department. of Health and Senior Services, United States; Miyagi Prefectural Institute of Public Health and Environment, Japan; Miyazaki Prefectural Institute for Public Health and Environment, Japan; Molde Hospital, Laboratory for Medical Microbiology, Norway; Molecular Diagnostics Unit , United Kingdom; Monash Medical Centre, Australia; Montana Laboratory Services Bureau, United States; Montana Public Health Laboratory, United States; Nagano City Health Center, Japan; Nagano Environmental Conservation Research Institute, Japan; Nagoya City Public Health Research Institute, Japan; Nara Prefectural Institute for Hygiene and

Environment, Japan; National Center for Communicable Diseases, Mongolia; National Center for Laboratory and Epidemiology, Laos; National Centre for Disease Control (NCDC), Mongolia; National Centre for Disease Control and Public Health, Georgia; National Centre for Preventive Medicine, Moldova, Republic of; National Centre for Scientific Services for Virology and Vector Borne Diseases, Fiji; National Health Laboratory, Japan; National Health Laboratory, Myanmar; National Influenza Center French Guiana and French Indies, French Guiana; National Influenza Center, Brazil; National Influenza Center, Mongolia; National Influenza Centre for Northern Greece, Greece; National Influenza Centre of Iraq, Iraq; National Influenza Lab, Tanzania, United Republic of; National Influenza Reference Laboratory, Nigeria; National Insitut of Hygien, Morocco; National Institute for Biological Standards and Control (NIBSC), United States; National Institute for Communicable Disease, South Africa; National Institute for Health and Welfare, Finland; National Institute of Health Research and Development, Indonesia; National Institute of Health, Korea, Republic of; National Institute of Health, Pakistan; National Institute of Hygien, Morocco; National Institute of Hygiene and Epidemiology, Vietnam; National Institute of Public Health - National Institute of Hygiene, Poland; National Institute of Public Health, Czech Republic; National Institute of Virology, India; National Microbiology Laboratory, Health Canada, Canada; National Public Health Institute of Slovakia, Slovakia; National Public Health Laboratory, Cambodia; National Public Health Laboratory, Ministry of Health, Singapore, Singapore; National Public Health Laboratory, Nepal; National Public Health Laboratory, Singapore; National Reference Laboratory, Kazakhstan; National University Hospital, Singapore; National Virology Laboratory, Center Microbiological Investigations, Kyrgyzstan; National Virus Reference Laboratory, Ireland; Naval Health Research Center, United States; Nebraska Public Health Lab, United States; Nevada State Health Laboratory, United States; New Hampshire Public Health Laboratories, United States; New Jersey Department of Health and Senior Services, United States; New Mexico Department of Health, United States; New York City Department of Health, United States; New York Medical College, United States; New York State Department of Health, United States; Nicosia General Hospital, Cyprus; Niigata City Institute of Public Health and Environment, Japan; Niigata Prefectural Institute of Public Health and Environmental Sciences, Japan; Niigata University, Japan; Nordlandssykehuset, Norway; North Carolina State Laboratory of Public Health, United States; North Dakota Department of Health, United States; Norwegian Institute of Public Health, Norway; Norwegian Institute of Public Health, Svalbard and Jan Mayen; Ohio Department of Health Laboratories, United States; Oita Prefectural Institute of Health and Environment, Japan; Okayama Prefectural Institute for Environmental Science and Public Health, Japan; Okinawa Prefectural Institute of Health and Environment, Japan; Oklahoma State Department of Health, United States; Ontario Agency for Health Protection and Promotion (OAHPP), Canada; Oregon Public Health Laboratory, United States; Osaka City Institute of Public Health and Environmental Sciences, Japan; Osaka Prefectural Institute of Public Health, Japan; Oslo University Hospital, Ulleval Hospital, Dept. of Microbiology, Norway; Ostfold Hospital - Fredrikstad, Dept. of Microbiology, Norway; Oswaldo Cruz Institute - FIOCRUZ - Laboratory of Respiratory Viruses and Measles (LVRS), Brazil; Papua New Guinea Institute of Medical Research, Papua New Guinea; Pasteur Institut of Cote d'Ivoire, Cote d'Ivoire; Pasteur Institute, Influenza Laboratory, Vietnam; Pathwest QE II Medical Centre, Australia; Pennsylvania Department of Health, United States; Prince of Wales Hospital, Australia; Princess Margaret Hospital for Children, Australia; Public Health Laboratory Services Branch, Centre for Health Protection, Hong Kong; Public Health Laboratory, Barbados; Puerto Rico Department of Health, Puerto Rico; Qasya Diagnostic Services Sdn Bhd, Brunei; Queensland Health Scientific Services, Australia; Refik Saydam National Public Health Agency, Turkey; Regent Seven Seas Cruises, United States; Royal Victoria Hospital, United Kingdom; Republic Institute for Health Protection, Macedonia, the former Yogoslav Republic of; Republic of Nauru Hospital, Nauru; Research Institute for Environmental Sciences and Public Health of Iwate Prefecture, Japan; Research Institute of Tropical Medicine, Philippines; Rhode Island Department of Health, United States; RIVM National Institute for Public Health and Environment, Netherlands; Robert-Koch-Institute, Germany; Royal Chidrens Hospital, Australia; Royal Darwin Hospital, Australia; Royal Hobart Hospital, Australia; Royal Melbourne Hospital, Australia; Russian Academy of Medical Sciences, Russian Federation; Rwanda Biomedical Center, National Reference Laboratory, Rwanda; Saga Prefectural Institute of Public Health and Pharmaceutical Research, Japan; Sagamihara City Laboratory of Public Health, Japan; Saitama City Institute of Health Science and Research, Japan; Saitama Institute of Public Health, Japan; Sakai City Institute of Public Health, Japan; San Antonio Metropolitan Health, United States; Sandringham,

National Institute for Communicable D, South Africa; Sapporo City Institute of Public Health, Japan; Scientific Institute of Public Health, Belgium; Seattle and King County Public Health Lab, United States; Sendai City Institute of Public Health, Japan; Servicio de Microbiología Clínica Universidad de Navarra, Spain; Servicio de Microbiología Complejo Hospitalario de Navarra, Spain; Servicio de Microbiología Hospital Central Universitario de Asturias, Spain; Servicio de Microbiología Hospital Donostia, Spain; Servicio de Microbiología Hospital Meixoeiro, Spain; Servicio de Microbiología Hospital Miguel Servet, Spain; Servicio de Microbiología Hospital Ramón y Cajal, Spain; Servicio de Microbiología Hospital San Pedro de Alcántara, Spain; Servicio de Microbiología Hospital Santa María Nai, Spain; Servicio de Microbiología Hospital Universitario de Gran Canaria Doctor Negrín, Spain; Servicio de Microbiología Hospital Universitario Son Espases, Spain; Servicio de Microbiología Hospital Virgen de la Arrixaca, Spain; Servicio de Microbiología Hospital Virgen de las Nieves, Spain; Servicio de Virosis Respiratorias INEI-ANLIS Carlos G. Malbran, Argentina; Shiga Prefectural Institute of Public Health, Japan; Shimane Prefectural Institute of Public Health and Environmental Science, Japan; Shizuoka City Institute of Environmental Sciences and Public Health , Japan; Shizuoka Institute of Environment and Hygiene, Japan; Singapore General Hospital, Singapore; Sorlandet Sykehus HF, Dept. of Medical Microbiology, Norway; South Carolina Department of Health, United States; South Dakota Public Health Lab, United States; Southern Nevada Public Health Lab, United States; Spokane Regional Health District, United States; St. Judes Childrens Research Hospital, United States; St. Olavs Hospital HF, Dept. of Medical Microbiology, Norway; State Agency, Infectology Center of Latvia, Latvia; State of Hawaii Department of Health, United States; State of Idaho Bureau of Laboratories, United States; State Research Center of Virology and Biotechnology Vector, Russian Federation; Statens Serum Institute, Denmark; Stavanger Universitetssykehus, Avd. for Medisinsk Mikrobiologi, Norway; Subdireccion General de Epidemiologia y Vigilancia de la Salud, Spain; Subdirección General de Epidemiología y Vigilancia de la Salud, Spain; Swedish Institute for Infectious Disease Control, Sweden; Swedish National Institute for Communicable Disease Control, Sweden; Taiwan CDC, Taiwan; Tan Tock Seng Hospital, Singapore; Tehran University of Medical Sciences, Iran; Tennessee Department of Health Laboratory-Nashville, United States; Texas Childrens Hospital, United States; Texas Department of State Health Services, United States; Thai National Influenza Center, Thailand; Thailand MOPH-U.S. CDC Collaboration (IEIP), Thailand; The Nebraska Medical Center, United States; Tochigi Prefectural Institute of Public Health and Environmental Science, Japan; Tokushima Prefectural Centre for Public Health and Environmental Sciences, Japan; Tokyo Metropolitan Institute of Public Health, Japan; Tottori Prefectural Institute of Public Health and Environmental Science, Japan; Toyama Institute of Health, Japan; U.S. Air Force School of Aerospace Medicine, United States; U.S. Naval Medical Research Unit No.3, Egypt; Uganda Virus Research Institute (UVRI), National Influenza Center, Uganda; Universidad de Valladolid, Spain; Università Cattolica del Sacro Cuore, Italy; Universitetssykehuset Nord-Norge HF, Norway; University Malaya, Malaysia; University of Florence, Italy; University of Genoa, Italy; University of Ghana, Ghana; University of Michigan SPH EPID, United States; University of Parma, Italy; University of Perugia, Italy; University of Pittsburgh Medical Center Microbiology Lab, United States; University of Sarajevo, Bosnia and Herzegovina; University of Sassari, Italy; University of the West Indies, Jamaica; University of Vienna, Austria; University of Virginia, Medical Labs/Microbiology, United States; University Teaching Hospital, Zambia; UPMC-CLB Dept of Microbiology, United States; US Army Medical Research Unit - Kenya (USAMRU-K), GEIS Human Influenza Program, Kenya; USAMC-AFRIMS Department of Virology, Cambodia; Utah Department of Health, United States; Utah Public Health Laboratory, United States; Utsunomiya City Institute of Public Health and Environment Science, Japan; VACSERA, Egypt; Vermont Department of Health Laboratory, United States; Victorian Infectious Diseases Reference Laboratory, Australia; Virginia Division of Consolidated Laboratories, United States; Wakayama City Institute of Public Health, Japan; Wakayama Prefectural Research Center of Environment and Public Health, Japan; Washington State Public Health Laboratory, United States; West Virginia Office of Laboratory Services, United States; Westchester County Department of Laboratories and Research, United States; Westmead Hospital, Australia; WHO National Influenza Centre Russian Federation, Russian Federation; WHO National Influenza Centre, National Institute of Medical Research (NIMR), Thailand; WHO National Influenza Centre, Norway; Wisconsin State Laboratory of Hygiene, United States; Wyoming Public Health Laboratory, United States; Yamagata Prefectural Institute of Public Health, Japan; Yamaguchi Prefectural Institute of Public Health and Environment, Japan;

Yamanashi Institute for Public Health, Japan; Yap State Hospital, Micronesia; Yokohama City Institute of Health, Japan; Yokosuka Institute of Public Health, Japan

