## [Decision Letter]

x

**Acceptance summary:**

This is a monumental effort to compare methods of predicting the evolution of the influenza A virus, the cause of seasonal flu, which is of critical importance for vaccine design. The study uses open-source models that integrate multiple types of genetic and phenotypic data about the virus and helps us understand why these models succeed or fail. A key discovery is that viral titers combined with sequence-based mutational load provide the best predictive power. Overall, the study highlights the sobering complexity of predicting H3 evolution in a variable and changing human immune landscape and the need for multiple strategies and forms of data integration to improve composite models going forward. This is interesting work for anyone interested in predicting evolution.

**Decision letter after peer review:**

Thank you for submitting your article "Integrating genotypes and phenotypes improves long-term forecasts of seasonal influenza A/H3N2 evolution" for consideration by *eLife*. Your article has been reviewed by two peer reviewers, and the evaluation has been overseen by a Reviewing Editor and George Perry as the Senior Editor. The reviewers have opted to remain anonymous.

The reviewers have discussed the reviews with one another and the Reviewing Editor has drafted this decision to help you prepare a revised submission.

We would like to draw your attention to changes in our revision policy that we have made in response to COVID-19 (https://elifesciences.org/articles/57162). Specifically, we are asking editors to accept without delay manuscripts, like yours, that they judge can stand as *eLife* papers without additional data, even if they feel that they would make the manuscript stronger. Thus the revisions requested below address clarity and presentation and would strengthen an already promising manuscript.

Summary:

This is a monumental effort to compare methods of predicting H3 evolution. The study demonstrates the value of composite, open-source models that integrate multiple types of genetic and phenotypic data. And it is particularly valuable for understanding the context under which different models and types of data perform better. Interestingly they find that HI titers combined with sequence-based mutational load provide the best predictive power. Along the way, the study also uncovers insights about the forces behind influenza evolution that are interesting in their own right. The exploration into why previously identified epitopes fail to predict modern patterns of evolution explains why HI is a better predictor. But it also highlights the flexibility of HA antigenicity and the volatility of the human immune landscape. Overall, the study highlights the sobering complexity of predicting H3 evolution in a variable and changing human immune landscape and the need for multiple strategies and forms of data integration to improve composite models going forward. This is interesting work for anyone interested in predicting evolution.

Essential revisions:

1) It would be helpful to further explore model performance in different contexts.

a) Overall, do the models perform any better or worse at predicting Northern v Southern hemisphere viruses populations?

b) I'm surprised the authors don't discuss more the recent problem of multiple H3 cocirculating clades (observed in Figure 9). Predicting H3 evolution has always been difficult, but at least there was generally a linear tree and a single dominant H3 clade at any given time. Did certain models better predict the emergence and persistence of this tree pattern? Do they offer insights?

2) Given that this study tackles a very real world problem (selecting strains for influenza vaccines), it would be helpful to have these results better translated for readers with public health backgrounds.

a) A simple addition would be an opening table/chart that describes the different models and categorizes them (lab data, sequence-based, tree-based etc.)

b) Figure 8 nicely visualizes model performance against actual vaccine strains. But can you quantify/summarize the results in this figure better (i.e., exactly how much 'closer' to the future than the vaccine strain?). Perhaps also including additional models in those measures?

c) In the Introduction can you provide more context for this study? What is the current range of influenza vaccine effectiveness? How frequent are H3 mismatches? And how well have we been trending at matching H3 vaccine strains to H3s in circulation? Are we making any discernible progress? Or have improvements in modeling been offset by the H3 cocirculating clades problem?

d) A key message from this study is how challenging prediction H3 is, even with new analysis tools and new types of experimental data. We have so much further to go. It's worth highlighting in the Discussion how CDC FluSight has collectively advanced epi flu forecasting by making weighted ensembles drawn from multiple modeling groups and how valuable a similar program would be for vaccine strain selection.

3) Some epitopes are more important than others. Did you consider weighting epitopes differently? Or would that just exacerbate overfitting? And how do you handle glycosylations in the model?

---

## [Author Response]

Essential revisions:1) It would be helpful to further explore model performance in different contexts.a) Overall, do the models perform any better or worse at predicting Northern v Southern hemisphere viruses populations?

We investigated whether our best model, HI antigenic novelty and mutational load, projected better estimates of the future for the Northern Hemisphere (forecasts made in October) or the Southern Hemisphere (forecasts made in April). To account for natural variation in distances between timepoints, we calculated the difference between the naive model’s distance to the future and our best model’s and plotted these values by Hemisphere (Author response image 1). The median adjusted distance to the future for the Northern and Southern Hemispheres were 0.45 and 0.53 AAs, respectively. These results suggest that there is not a substantial difference in our ability to forecast based on the Hemisphere.

**Author response image 1. sa2fig1:** Distribution of distances to the future by Hemisphere for the best natural model (HI antigenic novelty and mutational load) subtracted from the corresponding distance to the future for the naive model at the same timepoint.

b) I'm surprised the authors don't discuss more the recent problem of multiple H3 cocirculating clades (observed in Figure 9). Predicting H3 evolution has always been difficult, but at least there was generally a linear tree and a single dominant H3 clade at any given time. Did certain models better predict the emergence and persistence of this tree pattern? Do they offer insights?

We agree that this was an oversight on our part to not mention the unusual diversity of H3N2 clades in the last decade. We have modified the following paragraph in the Discussion to mention this pattern and its potential role in the performance of our models:

“Even the most accurate models with few parameters will sometimes fail due to the probabilistic nature of evolution. […] These results highlight the challenge of identifying models that remain robust to stochastic evolutionary events by avoiding overfitting to the past.”

As we note in this new text, we cannot disentangle model overfitting from changes in underlying evolutionary processes. This limitation unfortunately prevents us from identifying models that predicted the emergence or persistence of H3N2's current diversity. However, we can identify the models that are most robust to both overfitting and evolutionary change. We hope that this revised text clarifies these complexities and the reasons for selecting the HI antigenic novelty and mutational load model as our best model.

2) Given that this study tackles a very real world problem (selecting strains for influenza vaccines), it would be helpful to have these results better translated for readers with public health backgrounds.a) A simple addition would be an opening table/chart that describes the different models and categorizes them (lab data, sequence-based, tree-based etc.)

We have added a table to the first section of the Results (Table 1) as a summary of all primary models used with the simulated and natural populations. We reference this new table from the initial list of models in the same section (subsection “A distance-based model of seasonal influenza evolution”).

b) Figure 8 nicely visualizes model performance against actual vaccine strains. But can you quantify/summarize the results in this figure better (i.e., exactly how much 'closer' to the future than the vaccine strain?). Perhaps also including additional models in those measures?

To better represent and quantify how much closer to the future the observed and estimated closest strains are than the selected vaccine strains, we calculated the distance to the future of these strains relative to the vaccine strain at the corresponding timepoints and plotted these values in a new figure supplement (Figure 8—figure supplement 1). Strains with relative distances greater than zero were farther from the future than the vaccine strain. We have also added the closest strain selected by the naive model to Figure 8 and Figure 8—figure supplement 1, as an additional reference point for the strains selected by biologically-informed models.

We have updated the corresponding section of the Results to include a clearer description of how these closest strains are calculated, what they represent, and how much closer to the future each strain is than the vaccine strain:

“For each season when the WHO selected a new vaccine strain and one year of future data existed in our validation or test periods, we measured the observed distance of that strain’s sequence to the future and the corresponding distances to the future for the observed closest strains (Equation 3). […] These results were consistent with our earlier observations that the naive model often performs as well as biologically-informed models when estimating a single closest strain to the future.”

c) In the Introduction can you provide more context for this study? What is the current range of influenza vaccine effectiveness? How frequent are H3 mismatches? And how well have we been trending at matching H3 vaccine strains to H3s in circulation? Are we making any discernible progress? Or have improvements in modeling been offset by the H3 cocirculating clades problem?

We have modified the Introduction, splitting the first paragraph into two paragraphs (the first introducing seasonal influenza A/H3N2 and the second introducing predictive methods) and adding a new paragraph in between that summarizes vaccine effectiveness for H3N2 and factors affecting this effectiveness. This new paragraph is included below:

**“**Historically, the vaccine effectiveness of the H3N2 vaccine component has been much lower than the other seasonal influenza subtypes. […] While all of these factors must be addressed to increase vaccine effectiveness, substantial effort has focused on the selection of the most representative strain for the next season’s vaccine.”

d) A key message from this study is how challenging prediction H3 is, even with new analysis tools and new types of experimental data. We have so much further to go. It's worth highlighting in the Discussion how CDC FluSight has collectively advanced epi flu forecasting by making weighted ensembles drawn from multiple modeling groups and how valuable a similar program would be for vaccine strain selection.

We have updated the following sentence in the Discussion to explicitly mention the CDC’s FluSight network:

“The recent success of weighted ensembles for short-term influenza forecasting through the CDC’s FluSight network (Reich et al., 2019) suggests that long-term forecasting may benefit from a similar approach.”

3) Some epitopes are more important than others. Did you consider weighting epitopes differently? Or would that just exacerbate overfitting? And how do you handle glycosylations in the model?

Prior to our discovery of overfitting related to previously defined epitope sites, we had planned to test models fit to weighted epitope sites (for example, assigning more weight to Koel et al.’s seven sites than others). In light of model performance based on unweighted sites, we reasoned that weighting sites would most likely increase overfitting.

We chose not to fit models based on glycosylation sites based on two lines of reasoning. As these sites accumulate slowly in H3N2 relative to epitope mutations (six sites between 1968 and 2012), we expected there would not be a strong signal in year-to-year forecasts despite the biological importance of these events when they do occur. We also considered the results of Łuksza and Lässig’s original analysis which found that adding glycosylation sites did not improve their full fitness model and that these sites appeared correlated with epitope mutations.

As we briefly describe in the Discussion, we anticipate that antigenic escape assays like those presented in Lee et al., 2019, could allow us to identify and weigh contemporary epitope sites. How to effectively translate these antigenic escape data to fitness metrics is an active area of research in the Bedford lab.